# Independent regulation of Z-lines and M-lines during sarcomere assembly in cardiac myocytes revealed by the automatic image analysis software sarcApp

**Abigail C Neininger-Castro[1]\*, James B Hayes[1], Zachary C Sanchez[1], Nilay Taneja[1], Aidan M Fenix[1], Satish Moparthi[2], Stéphane Vassilopoulos[2], Dylan Tyler Burnette[1]\***

[1]Department of Cell and Developmental Biology, Vanderbilt University School of Medicine Basic Sciences, Nashville, United States; [2]Sorbonne Université, INSERM, Institut de Myologie, Centre de Recherche en Myologie, Paris, France

**\*For correspondence:** abbieneininger@gmail.com (ACN-C); dylan.burnette@vanderbilt.edu (DTylerB)

**Competing interest:** The authors declare that no competing interests exist.

**Abstract** Sarcomeres are the basic contractile units within cardiac myocytes, and the collective shortening of sarcomeres aligned along myofibrils generates the force driving the heartbeat. The alignment of the individual sarcomeres is important for proper force generation, and misaligned sarcomeres are associated with diseases, including cardiomyopathies and COVID-19. The actin bundling protein, α-actinin-2, localizes to the 'Z-Bodies" of sarcomere precursors and the 'Z-Lines' of sarcomeres, and has been used previously to assess sarcomere assembly and maintenance. Previous measurements of α-actinin-2 organization have been largely accomplished manually, which is time-consuming and has hampered research progress. Here, we introduce sarcApp, an image analysis tool that quantifies several components of the cardiac sarcomere and their alignment in muscle cells and tissue. We first developed sarcApp to utilize deep learning-based segmentation and real space quantification to measure α-actinin-2 structures and determine the organization of both precursors and sarcomeres/myofibrils. We then expanded sarcApp to analyze 'M-Lines' using the localization of myomesin and a protein that connects the Z-Lines to the M-Line (titin). sarcApp produces 33 distinct measurements per cell and 24 per myofibril that allow for precise quantification of changes in sarcomeres, myofibrils, and their precursors. We validated this system with perturbations to sarcomere assembly. We found perturbations that affected Z-Lines and M-Lines differently, suggesting that they may be regulated independently during sarcomere assembly.

## eLife assessment

This manuscript describes a **useful** tool for quantitative assessment of sarcomere structures in healthy and perturbed cardiomyocytes grown in vitro. The work is **solid**, and the methods, data and analyses broadly support the claims with only minor weaknesses. The tool will be relevant to biologists working on and interested in obtaining quantitative information on sarcomere structure, function and development.

## Introduction

The sarcomere is the fundamental unit of contraction in striated muscle (*Au, 2004*). Sarcomeres individually contract to generate a force, and sarcomeres organize and align within myofibrils to augment that force. These forces enable skeletal muscle to coordinate organism-level locomotion and cardiac muscle to drive the heartbeat (*Hersch et al., 2013*; *Wang et al., 2021*). Sarcomere structure and alignment is critical to muscle function, and dysfunctional sarcomeres and myofibrils have been implicated as causes of disease (*Eschenhagen et al., 2015*; *Harvey and Leinwand, 2011*; *Olsson et al., 2004*).

Since the mid-20th century, the question of how sarcomeres are assembled and subsequently organized into higher-order myofibrils has been a much-debated topic, with multiple separate lines of research emerging to support a small host of potential models (*Dlugosz et al., 1984*; *Sanger et al., 1984*; *McKenna et al., 1985*). Generally, the models of sarcomere assembly that are backed by the most data posit that sarcomeres arise from either (i) a 'parts-wise' assembly, where precursors of unique sarcomeric regions (e.g., the thin [actin] or thick [myosin] filament) are first assembled separately, then 'stitched' together to generate the final structure (i.e., Stitching Model) or (ii) a more direct assembly, where a precursor forms that resembles a sarcomere but has a subset of non-sarcomeric proteins that are later replaced by sarcomeric proteins (i.e., Pre-myofibril Model) (*Dlugosz et al., 1984*; *Rhee et al., 1994*). Recently, our group has shown that sarcomeres descend directly from muscle stress fibers (MSFs), with our data supporting a subset of predictions made by both models, but not necessarily favoring either model (*Fenix et al., 2018*). All models of sarcomere assembly are currently limited in their testability because the field lacks a tractable workflow for high-throughput analysis.

Recently, our lab has shown that human induced pluripotent stem cell-derived cardiac myocytes (hiCMs) can be harnessed as a tractable model for routine imaging of sarcomere assembly in live CMs (*Fenix et al., 2018*; *Fenix et al., 2016*). In our assay, hiCMs assemble sarcomeres de novo, and directly from MSFs, within 24 hr. Despite having powerful tools for imaging, our analysis workflow is still hampered by technical bottlenecks related to (i) image binarization and (ii) manual quantification. Binarization via classical methods relying on image pixel intensity (e.g., Otsu's method) is often performed manually – slowly – and can become muddled by a multitude of factors, including (i) high-intensity artifacts related to nonspecific localization of antibodies, (ii) low-intensity staining due to high background levels/noise, and/or (iii) multi-valent interactions by the target protein itself occluding the specific structure of interest (*Otsu, 1979*; *Sankur, 2004*; *Uchida, 2013*). Meanwhile, we find that even well-binarized, high-resolution images can require several hours of manual quantification per image.

Deep learning has emerged as a powerful potential solution for rapid, accurate binarization of complex grayscale images. Deep learning-based binarization can be accomplished using a framework called a U-Net (*Ronneberger and Brox, 2015*). A U-Net trains a deep learning model to convert images into only background (noise) and foreground (signal) by matching images to expert-annotated 'ground truth' binaries – that is, images manually annotated by the user. Compared to manually performed binarization, U-Net offers the advantages of superior accuracy and speed, but at a high upfront cost, since implementation of a U-Net requires a user with advanced background in mathematics, experience with machine learning, and proficiency in writing software code (*Ronneberger and Brox, 2015*; *Yin et al., 2022*).

While a sarcomere contains 100+ proteins, the field has often resorted to 'proxying' sarcomere assembly by staining, often exclusively, for α-actinin-2 (*Fenix et al., 2018*; *Henderson et al., 2017*; *Chopra et al., 2018*; *Ehler et al., 1999*; *Hinson et al., 2015*; *Taneja et al., 2020*). α-actinin-2 marks the sarcomeric Z-Lines, which border the sarcomeric contractile machinery (*Murphy and Young, 2015*; *Ribeiro et al., 2014*). In the last decade, several methods for automatic quantification of Z-Lines have emerged, with the most widely used methods relying on calculations made in frequency space (i.e., by using a Fourier transformation of the image) (*Chopra et al., 2018*; *Hinson et al., 2015*). While frequency space calculations can detect repeating patterns created by adjacent Z-Lines, current methods yield no information about Z-Lines in real space (e.g., number, size, spacing, or organization within myofibrils), about sarcomere precursors, or about any sarcomeric components other than Z-Lines. Moreover, it is not yet clear quite how the assembly of Z-Lines themselves relates either spatially or temporally to the assembly of other sarcomere components, or vice versa.

Here, we present a high-throughput, automated, and non-biased quantification scheme for sarcomere assembly. Our approach is two-pronged and includes a unique method for both (1) rapid, accurate binarization of images that can consistently separate true signal from noise and structures of interest from those that are not of interest and (2) automated quantification of relevant sarcomeric and pre-sarcomeric structures within the binarized images. Towards (1), we have developed 'yoU-Net,' a U-Net-based framework with a user-friendly graphical user interface (GUI) that enables a user with little to no experience to binarize images with the power and flexibility afforded by deep learning. We demonstrate that yoU-Net can accurately binarize images of CM sarcomeres and sarcomere precursors using multiple unique sarcomere stains. Towards (2), we have developed 'sarcApp,' a software that automatically annotates and calculates 50+ unique descriptive outputs of sarcomeres and sarcomere precursors in real space using binarized images. Using yoU-Net and sarcApp, we demonstrate that some perturbations to sarcomere assembly preferentially affect specific sarcomere components (e.g., M-Lines) more than others (e.g., Z-Lines). Our data emphasize the need for more high-throughput studies into sarcomere assembly that includes multiple sarcomere components, and altogether point to a disjointed or modular sarcomere assembly program within CMs that is more nuanced than was previously believed. The tools presented herein will facilitate future high-throughput studies to develop a more mature and complex model of sarcomere assembly.

## Results

### yoU-Net enables user-friendly binarization of dynamic grayscale images

Our overarching goal was to develop a method for fast and accurate quantification of sarcomeres within hiCMs using our sarcomere assembly assay. We faced two primary bottlenecks in our workflow, the first of which was a slow and cumbersome image binarization process. Binarization facilitates image quantification by converting the complex, raw image of a biological specimen into a far simpler image containing only background (non-relevant information) and signal (relevant information). A well-binarized image should retain high fidelity to the original image but contain only the relevant structures of interest superimposed over a blank background. Individual pixels within a binary image are stripped of dynamic information present within the raw image and are re-assigned to represent only either background or signal – binary images are therefore often represented in black (background) and white (signal).

Traditional methods of binarization fall short for images of sarcomeres and sarcomere precursors in hiCMs due to the dynamic range of pixel intensities present within the raw images. In images of hiCMs 24 hr after plating, sarcomere precursors and sarcomeres are each stained by α-actinin-2 (*Figure 1A*), but the Z-Lines of the sarcomeres are markedly brighter than the Z-Bodies of the precursor structures (i.e., MSFs). Classical, intensity-based binarization methods force a user to make a choice in this scenario – specifically, whether to accurately binarize based on gray levels of the sarcomere precursors or of the sarcomeres themselves (without the consistent option for both). Classical Otsu-based thresholding (using FIJI) of a representative cell in *Figure 1A* shows that binarizing this cell according to Z-Lines oversaturates Z-Bodies into an inseparable clump (*Figure 1A*, arrowhead), leaving a user unable to accurately quantify both structures from this single binary.

Our lab was well-positioned to approach this problem by constructing a deep learning-based model called a U-Net (*Ronneberger and Brox, 2015*). A U-Net enables a user to train a deep learning model to automatically binarize images with high precision and speed. Model 'training' involves matching raw images to user-annotated binaries – called 'ground truth' binaries – using a complex set of mathematical convolution/downsampling and deconvolution/upsampling steps. The architectural layout of a U-Net is represented by *Figure 1B*.

We constructed our own U-Net and trained it using 1000 training steps (100 epochs/steps of 10 iterations each) of hiCMs stained for α-actinin-2. After training, our U-Net can reliably and automatically predict high-fidelity binaries of hiCM α-actinin-2 stains that are resistant to artifacts related to differences in pixel intensity and can accurately separate individual Z-Bodies (*Figure 1C*, arrowhead). Recognizing that other labs would benefit from the flexibility and accuracy of our U-Net, we also wrote software encoding for a GUI to accompany our U-Net in the hopes of making it more user-friendly and broadly accessible. Our U-Net and GUI together are known as 'yoU-Net' and are available, open

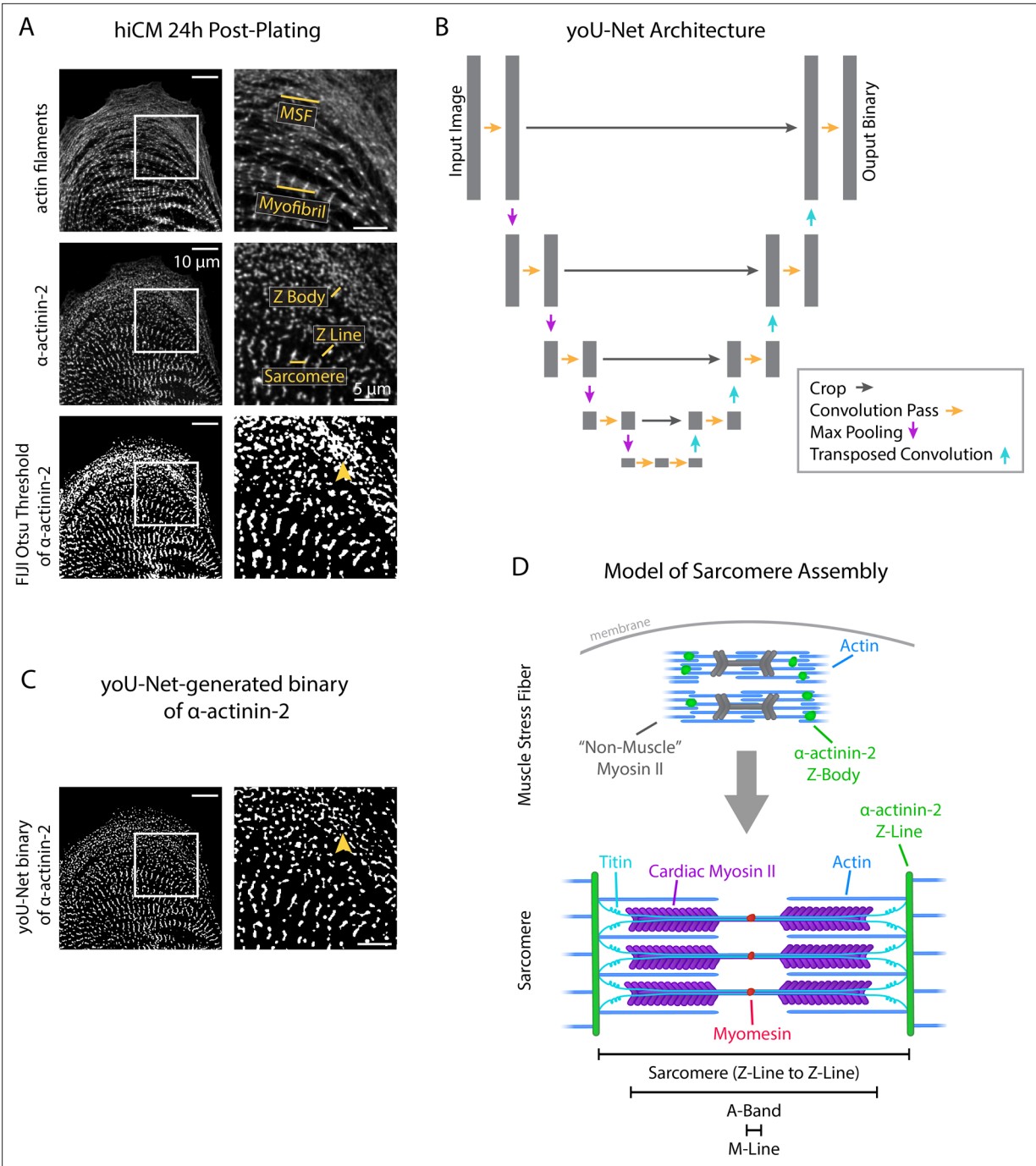

**Figure 1.** yoU-Net follows a U-Net architecture to binarize immunofluorescence images. (**A**) Representative images of α-actinin-2 and actin filaments (phalloidin) in a human induced pluripotent stem cell-derived cardiac myocyte (hiCM). The α-actinin-2 binary was thresholded in FIJI using Otsu's method. Orange arrowhead denotes Z-Bodies. (**B**) yoU-Net architecture from input image to output binary. Details can be found in the Methods and in *Figure 1—figure supplement 1*. (**C**) yoU-Net-generated binary of α-actinin-2, predicted using the trained U-Net described in (**B**). Orange arrowhead: Z-Bodies. (**D**) Model of muscle stress fibers (MSFs) and myofibrils during sarcomere formation. Black arrow denotes direction of MSF translocation as α-actinin-2-positive Z-Bodies elongate and coalesce to form Z-Lines.

The online version of this article includes the following figure supplement(s) for figure 1:

**Figure supplement 1.** Details of yoU-Net architecture and trained model generation.

source with details in the supplement (*Figure 1—figure supplement 1*). We used yoU-Net to binarize images throughout the remainder of this study.

## sarcApp quantifies muscle stress fibers, sarcomeres, and myofibrils

The next bottleneck within our workflow was the issue of manual quantification, which in some cases can require several hours for a single CM. A meaningful quantitative output of sarcomere assembly would discriminate between sarcomeres and sarcomere precursors (MSFs), and produce readouts for each. Sarcomere assembly involves the formation MSFs near the cell edge that give rise to sarcomeres typically closer to the cell center (*Figure 1D*). Within MSFs are punctate α-actinin-2-containing structures termed 'Z-Bodies,' which gradually elongate/concatenate over time and eventually transition into the Z-Lines of sarcomeres (*Figure 1D*). While the Z-Body to Z-Line transition exists along a continuum, it was necessary for us to define a binary transition point on the basis of α-actinin-2 structure length when a Z-Body 'becomes' a Z-Line. We tasked individual lab members with independently annotating 10 images each to determine which structures constituted a Z-Body and which were Z-Lines. Together, we defined a potential Z-Line to be any α-actinin-2-containing structure with a length >1.4 μm, with all other structures being potential Z-Bodies. Because many organisms have variable sarcomere spacing and structure, this variable along with all others discussed (lengths, minimum number of structures, etc.) are customizable and can be easily redefined by the user for their unique model system (*Reedy and Beall, 1993*).

Using our predefined size criterion for Z-Lines and Z-Bodies, we next desired a method capable of automated detection, measurement, and descriptive output of both structures. Towards this goal, we developed sarcApp, a software-based code that classifies each α-actinin-2 structure within a binary image as a Z-Line (*Figure 2A and B*) or Z-Body (*Figure 2C and D*) and further assigns each to either a myofibril (*Figure 2B*) or MSF (*Figure 2D*). All structures identified as potential Z-Lines are paired to other potential Z-Lines in the cell based on shape, orientation, and location, then assigned to myofibril based on orientation to other Z-Lines (*Figure 2A and B*). Z-Lines, which belong to a myofibril, are considered 'confirmed.' This eliminates singular α-actinin-2-positive adhesions. Next, each potential Z-Body is paired to other Z-Bodies based on shape and location, then assigned to an MSF based on orientation to other nearby Z-Bodies (*Figure 2C and D*). Like with Z-Lines, only Z-Bodies that belong to an MSF are considered 'confirmed.' After detecting, annotating/assigning, and measuring all potential Z-Lines and Z-Bodies within a given cell, sarcApp automatically compiles and generates three spreadsheets summarizing the data on a per-cell, per-myofibril, and per-MSF basis. Cell spreadsheets generated include myofibrils per cell (*Figure 2E*), Z-Lines per cell (*Figure 2F*), Z-Line lengths (*Figure 2G*), myofibril persistence lengths (*Figure 2H*), MSFs per cell (*Figure 2I*), Z-Bodies per cell (*Figure 2J*), Z-Body lengths (*Figure 2K*), and MSF persistence lengths (*Figure 2L*). Myofibril spreadsheets include analysis of myofibril organization within the cell by calculating the angle of each myofibril relative to the nearest cell edge (*Figure 2M–O*), as well as number of Z-Lines, average Z-Line spacing, persistence length, absolute angle of the myofibril long axis (considering the image edges as the X and Y axes), average Z-Line length, distance from the center point of the myofibril to the nearest parallel cell edge, absolute angle of the nearest cell edge, and relative angle of the myofibril to the edge (a measure of parallelism); all on a per individual myofibril basis. MSF spreadsheets include number of Z-Bodies, average Z-Body spacing, and persistence length for each MSF.

sarcApp must take certain liberties to parameterize and predict MSF/myofibrillar structures within CMs. For example, Z-Lines are assigned to individual myofibrils in part by inter-Z-Line distance (which we define as being <3 μm apart using centroid-to-centroid distance) but also by Z-Line long-axis angle (which we define as being within 30°, accounting for cell curvature). These measurements are customizable as some species have varied inter-Z-Line distance or varied cell curvature. We increase the computational speed of sarcApp by measuring Z-Line spacing using the distance between each Z-Line's centroid as opposed to parallel spacing, which can produce slightly different results (*Figure 2—figure supplement 1A*). sarcApp defines a myofibril as a structure with a linear collection of four or more Z-Lines (i.e., three sarcomeres). Potential Z-Lines that are not assigned to a myofibril are not quantified, as they could represent adhesions, which are also α-actinin-2-positive (*Taneja et al., 2020*). Any structure with a long axis <1.4 μm is defined as a potential Z-Body. Z-Bodies are assigned to MSFs with the criteria that the Z-Bodies are (1) <3 μm apart and (2) have centroids that

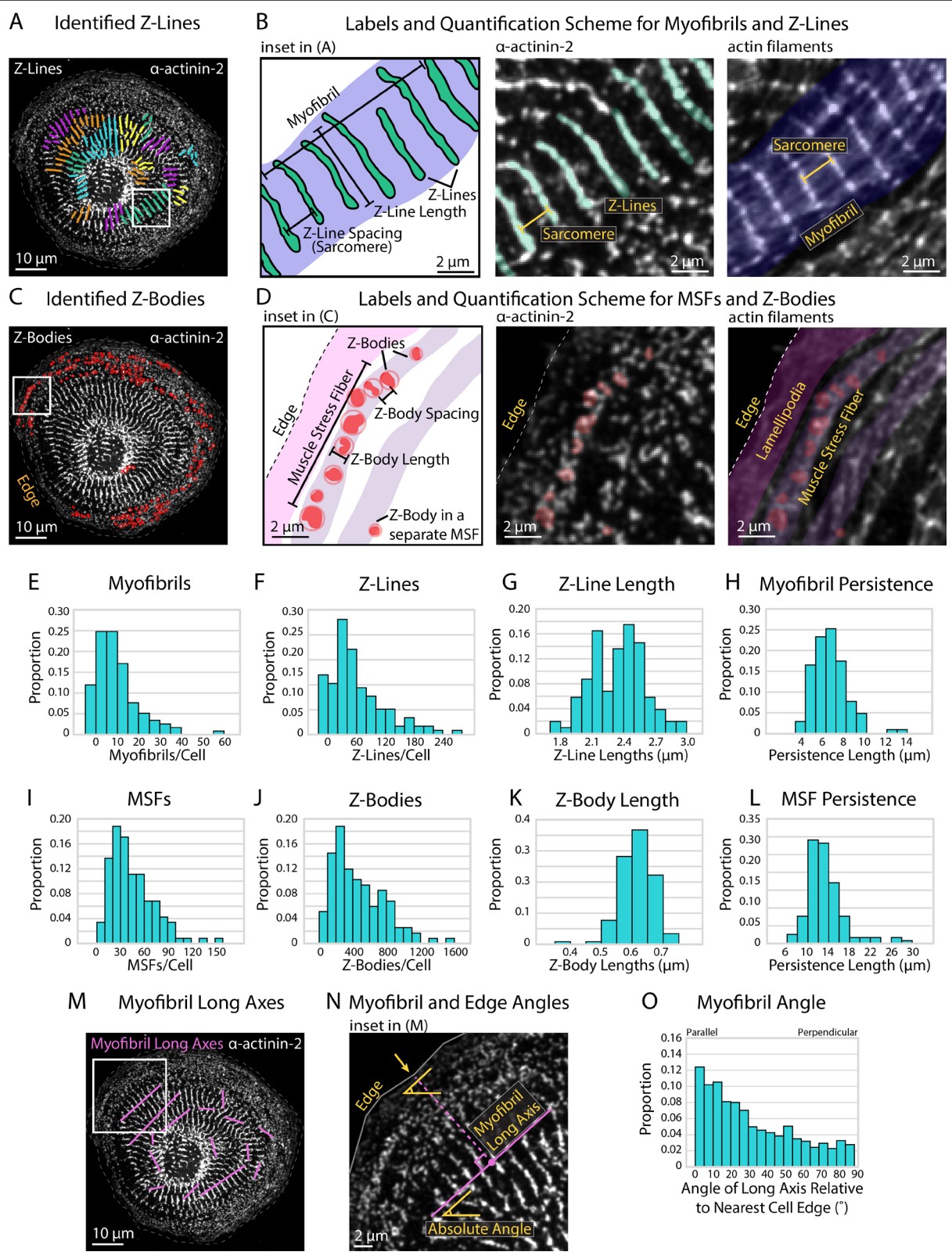

**Figure 2.** Quantifying sarcomere and myofibril organization using α-actinin-2 binaries. (**A**) Z-Lines and myofibrils identified by sarcApp. Each line denotes a Z-Line, and each different color represents a different myofibril. (**B**) Quantification scheme for myofibrils and Z-Lines. Details can be found in *Figure 2—figure supplement 1*. (**C**) Z-Bodies identified by sarcApp. Each red circle denotes a Z-Body. (**D**) Quantification scheme for muscle stress fibers (MSFs) and Z-Bodies. Details can be found in *Figure 2—figure supplement 1*. (**E**) Distribution of myofibrils per human induced pluripotent

*Figure 2 continued on next page*

*Figure 2 continued*

stem cell-derived cardiac myocyte (hiCM) plated for 24 hr (N = 188 cells; four biological replicates). (**F**) Distributions of Z-Lines per hiCMs from (**E**). (**G**) Distribution of average Z-Line length per cell from (**F**) (N = 104 cells). (**H**) Distribution of average myofibril persistence lengths per cell from (**G**) (N = 104 cells; quantification details can be found in *Figure 2—figure supplement 1*). (**I**) Distribution of MSFs per cell from (**E**). (**J**) Distribution of Z-Bodies per cell from (**E**). (**K**) Distribution of average Z-Body length per cell from (**E**). (**L**) Distribution of average MSF persistence lengths per cell from (**E**). (**M**) Myofibril long axes identified by sarcApp. (**N**) Quantification scheme for myofibril angle relative to edge. Briefly, the closest edge segment to the myofibril long axis (perpendicularly) is used as the reference angle, and the numerical output is the difference between the myofibril long axis angle and the reference edge angle. (**O**) Distribution of myofibril orientation relative to cell edge in the same cells as (**E**) (N = 188 cells, 1217 myofibrils). Note that most myofibrils are relatively parallel to the edge in hiCMs plated for 24 hr.

The online version of this article includes the following figure supplement(s) for figure 2:

**Figure supplement 1.** Details of geometric calculations used in sarcApp.

form a line within 30° parallel to the edge, indicative of a typical Z-Body organization with MSF actin filaments (*Fenix et al., 2018*).

## Blebbistatin treatment reduces Z-Line assembly in hiCMs

To validate the functionality and accuracy of sarcApp, we paired our lab's sarcomere assembly assay (*Fenix et al., 2018*; *Taneja et al., 2020*) with a well-established inhibitor of sarcomere assembly: Blebbistatin, a pan-myosin II inhibitor (*Limouze et al., 2004*). In brief, we trypsinized and replated hiCMs. During this process, the hiCMs lose and reform their sarcomeres, allowing us to monitor sarcomere assembly. Because this assembly begins at the cell edge and moves rearward toward the cell center, a snapshot of a hiCM after 24 hr of spreading (once sarcomeres have begun to form and myofibrils begin to align) contains information on early, mid, and late sarcomere formation at the leading edge, behind the edge in the lamella, and toward the center of the cell, respectively (*Fenix et al., 2018*).

Blebbistatin inhibits myosin II-based contractility and Blebbistatin-treated hiCMs exhibit reduced Z-Line assembly in our assay (*Figure 3A*, *Figure 3—figure supplement 1*), consistent with other reports (*Chopra et al., 2018*; *Liu et al., 2016*). Platinum-replica electron microscopy (EM) shows that Blebbistatin-treated hiCMs do not appear to have well-defined Z-Lines and have a disordered overall orientation of MSFs/myofibrils compared to controls (*Figure 3B*). As measured by sarcApp, hiCMs treated with Blebbistatin exhibited fewer myofibrils per cell (*Figure 3C*), fewer Z-Lines per cell (*Figure 3D*), decreased myofibril persistence length (*Figure 3E*), decreased Z-Line length (*Figure 3F*), and decreased overall size of all α-actinin-2-positive puncta/structures (*Figure 3G*). sarcApp also reported that, while the myofibrils of control hiCMs are oriented approximately parallel to the nearest cell edge (*Figure 3H*; similar to *Figure 2M–O*), the myofibrils of Blebbistatin-treated hiCMs are more randomly oriented (*Figure 3H*), suggesting a disorganized assembly process. Altogether, these data generated by sarcApp are consistent with a wide range of previous studies showing that Blebbistatin inhibits sarcomere assembly (*Chopra et al., 2018*; *Liu et al., 2016*).

## sarcApp quantifies titin in hiCMs

Our next goal was to introduce additional quantitative modes of analysis to sarcApp that could be used concurrently with α-actinin-2-based quantification of Z-line assembly. Our focus was on other core components of the sarcomere which, like α-actinin-2, are thought to play an active role in assembly and can be stained and visualized using commercially available antibodies. Our group has shown previously that the protein titin can be used as a spatiotemporal indicator of myofibril maturation status in hiCMs (*Taneja et al., 2020*). Antibodies to the titin I-band localize to both MSFs and sarcomeres in hiCMs (*Figure 4A*). Titin forms ringlike structures around the Z-Bodies of MSFs that are closer to the apparent sarcomere transition point (*Figure 4A*). *Figure 4B* shows our current model for how titin is oriented around Z-bodies during earlier stages of sarcomere assembly, forming rings. Likely, due to both antibody localization to the titin I-Band region and the ability of titin to bind both actin and the α-actinin-2 N-terminus, titin is oriented with the N-terminus toward the center of the Z-Body and the C-terminus radially oriented outward (*Figure 4C*; *Anderson and Granzier, 2012*; *Herzog, 2018*). This is a topic of current research in our lab.

yoU-Net-generated binaries of titin stains after training with annotated ground-truth binaries accurately recapitulate titin rings (MSFs) and titin doublets (sarcomeres) in hiCMs (*Figure 4A*, *Figure 1—figure supplement 1D*). We parameterized each titin-based structure into a quantitative scheme and

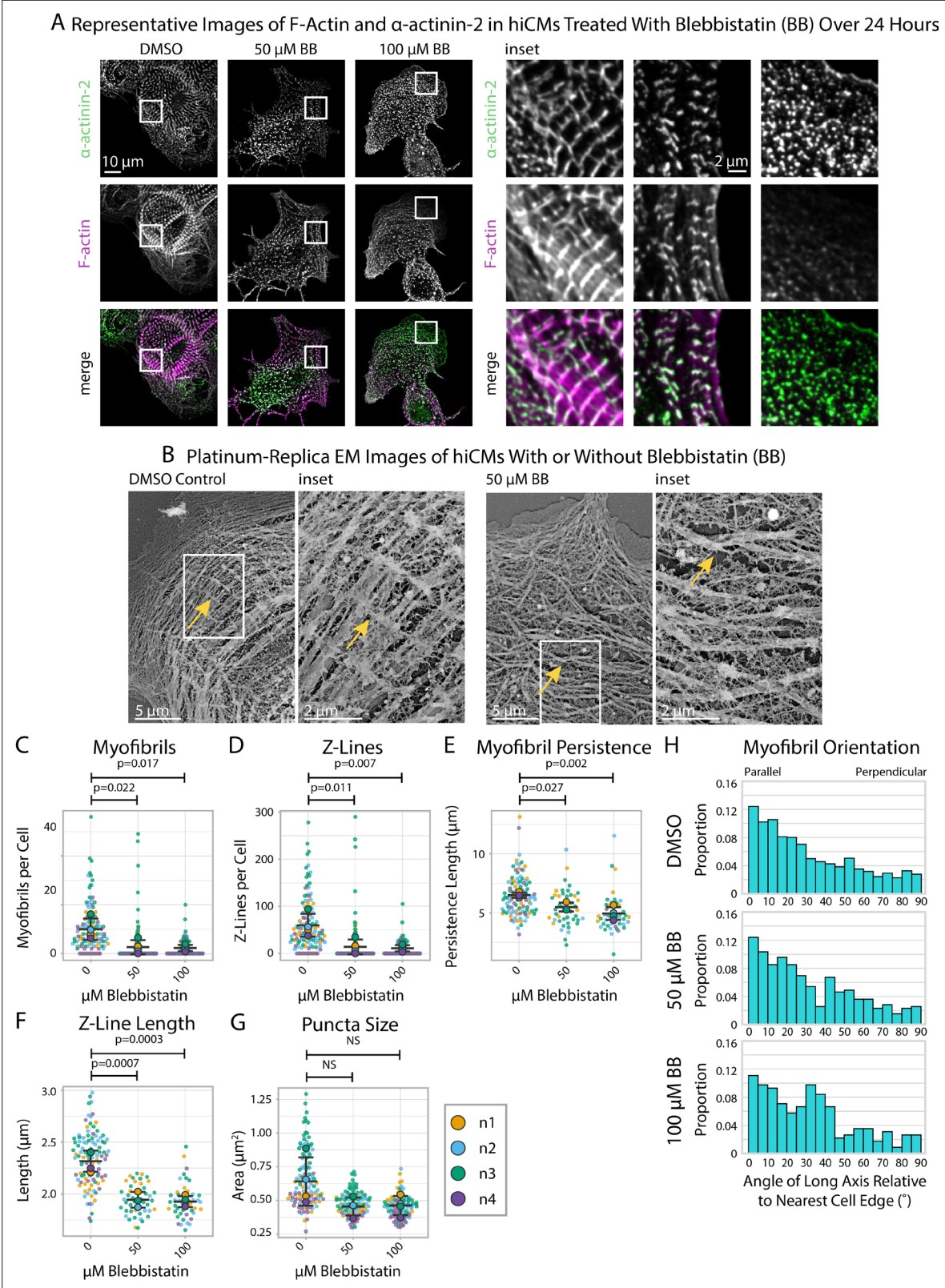

**Figure 3.** Blebbistatin treatment ablates Z-Line formation in human induced pluripotent stem cell-derived cardiac myocytes (hiCMs). (**A**) Representative images and insets of α-actinin-2 and F-actin in hiCMs treated with DMSO, 50 µM Blebbistatin, or 100 µM Blebbistatin. (**B**) Representative platinum replica EM image of a control hiCM and a 50 µM Blebbistatin-treated hiCM. Arrows indicate an elongated Z-Line in the DMSO-treated hiCM and a Z-Body in the Blebbistatin-treated hiCM. (**C**) Myofibrils per cell in hiCMs (N = 118 DMSO cells, 108 50 µM Blebbistatin cells, and 93 100 µM Blebbistatin

*Figure 3 continued on next page*

*Figure 3 continued*

cells; four biological replicates). (**D**) Z-Lines per cell in hiCMs from (**C**). (**E**) Average myofibril persistence length per cell in hiCMs from (**C**) (N = 104 DMSO control cells, 45 50 μM Blebbistatin cells, and 45 100 μM Blebbistatin cells). (**F**) Average Z-Line length per cell from (**E**). (**G**) Average size of all α-actinin-2-positive puncta in hiCMs from (**C**). (**H**) Myofibril orientation relative to the cell edge segment closest to the myofibril center, perpendicularly. N = 4 biological replicates, 1217 DMSO control myofibrils, 385 50 μM Blebbistatin myofibrils, 220 100 μM Blebbistatin myofibrils. Blebbistatin treatment for 6 and 12 hr can be found in *Figure 3—figure supplement 2* and *Figure 3—figure supplement 3*.

The online version of this article includes the following figure supplement(s) for figure 3:

**Figure supplement 1.** α-actinin-2 quantification and organization in Blebbistatin-treated human induced pluripotent stem cell-derived cardiac myocytes (hiCMs) 6 hr post-plating.

**Figure supplement 2.** α-actinin-2 quantification and organization in Blebbistatin-treated human induced pluripotent stem cell-derived cardiac myocytes (hiCMs) 12 hr post-plating.

**Figure supplement 3.** α-actinin-2 quantification and organization in Blebbistatin-treated human induced pluripotent stem cell-derived cardiac myocytes (hiCMs) 24 hr post-plating.

have outfitted sarcApp with a modality for measuring titin. To determine the minimum length required for a potential titin doublet to be considered as part of a sarcomere, we co-stained hiCMs with both titin and α-actinin-2, and found that on average, titin doublets are 0.3 μm longer than the Z-Lines with which they associate. Thus, we set a lower boundary of titin doublet length at 1.7 μm (Z-lines: 1.4 μm). Closer inspection of titin images revealed that, in several cases, the titin I-Band signal 'wraps' around the radial tip of Z-Lines, reminiscent of titin rings around Z-Bodies.

sarcApp accurately distinguishes between titin doublets (*Figure 4C*) and titin rings (*Figure 4E*) and produces readouts for number of myofibrils doublets per cell, myofibril persistence length, doublet length, number of rings, ring diameter, ring aspect ratio, distance of doublets from the edge, and distance of rings from the edge. (*Figure 4C–F*). sarcApp is additionally equipped to incorporate a cell edge co-stain (e.g., actin or non-muscle myosin IIA or IIB) alongside titin to measure myofibril orientation and myofibril distance from edge since titin stains do not label the edge (similar to *Figure 2M–O*).

## Blebbistatin affects myofibril orientation and doublet length, but not the periodicity of titin structures

We wished to use our sarcomere formation assay in the presence of Blebbistatin to quantify titin structure alignment using sarcApp. As before, hiCMs were replated in the presence of Blebbistatin for 24 hr, then fixed and stained for titin as well as phalloidin (F-actin) to aid in the visualization of MSFs. While F-actin in Blebbistatin-treated hiCMs is mostly punctate, titin rings and doublets can still be observed by eye around punctate actin with some degree of organization and periodicity (*Figure 5A*). sarcApp-dependent quantification of myofibrils using titin revealed no significant differences in the numbers of myofibrils in hiCMs replated in moderate (50 μM) or high (100 μM) Blebbistatin (*Figure 5B*), but found fewer and shorter titin doublets in high Blebbistatin (*Figure 5C and D*). sarcApp-based quantification of titin precursor rings within MSFs revealed no difference in absolute number of rings per cell with Blebbistatin treatment (*Figure 5E*) but rings that were detected were significantly more rounded as indicated by a decreased aspect ratio (*Figure 5F*). sarcApp also detected that myofibrils as quantified by titin were mostly parallel to the edge in control cells but became more randomly aligned relative to the edge in Blebbistatin (*Figure 5G*), consistent with sarcApp-dependent quantification of α-actinin-2. In summary, Blebbistatin treatment results in disarrayed and shorter titin doublets that are still capable of aligning into periodic myofibrillar structures.

## sarcApp quantifies myomesin, a component of the M-Line, in hiCMs

Having equipped sarcApp to quantify two structures associated with Z-Line assembly in hiCMs, our next goal was to introduce a modality for M-Line assembly. The M-Line lies at the midline of the stack of myosin II filaments in the sarcomere and demarcates the midpoint between two Z-Lines (*Figure 6A*). M-Lines in hiCMs can be stained using antibodies to the protein myomesin. Myomesin stains closely resemble α-actinin-2 Z-Line stains although myomesin is apparently absent from Z-Bodies (*Figure 6B*). Like α-actinin-2 and titin stains, yoU-Net can be used to predict myomesin binaries with high accuracy (*Figure 6B*, *Figure 1—figure supplement 1E*).

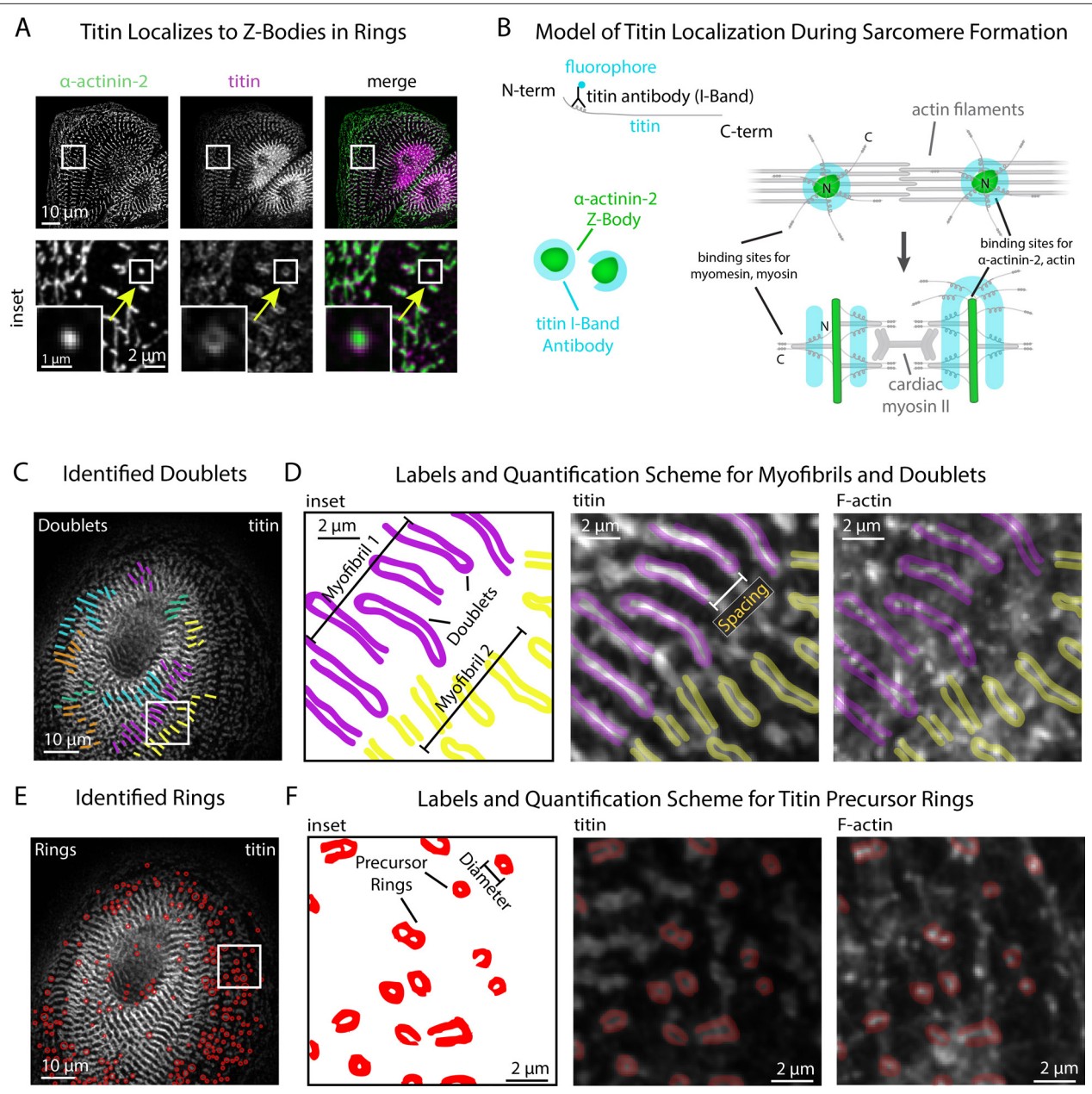

**Figure 4.** sarcApp uses titin binaries to identify myofibrils and precursor ring structures. (**A**) Representative image of titin and α-actinin-2 in a control human induced pluripotent stem cell-derived cardiac myocyte (hiCM). Arrow: an α-actinin-2-positive Z-Body with titin localized in a ring around it. (**B**) Model of titin localization during sarcomere formation. (**C**) Titin doublets identified by sarcApp. Each line denotes a doublet with titin localized, and each color is a myofibril. (**D**) Quantification scheme for myofibrils and titin doublets. Details can be found in *Figure 2—figure supplement 1* and the Methods. (**E**) Titin precursor rings identified by sarcApp (red). (**F**) Quantification scheme for titin precursor rings. Details can be found in *Figure 2—figure supplement 1* and the Supplemental Methods.

We equipped sarcApp to detect and measure myomesin M-Lines from yoU-Net-generated binaries (*Figure 6C*) based on a scheme that resembles the one designed for Z-Lines (*Figure 6D*). For myomesin, the M-Line must >1.4 μm and adjacent to at least two other M-lines to be considered part of a myofibril (i.e., at least three M-lines per myofibril, as it could be inferred that such a myofibril would have at least four Z-lines). sarcApp-defined readouts using myomesin include myofibrils per cell, M-Lines per cell, myofibril persistence length, and M-Line lengths. As with titin, sarcApp is additionally equipped to incorporate a cell edge co-stain (e.g., actin or non-muscle myosin IIA or IIB) alongside myomesin to measure myofibril orientation and myofibril distance from cell edge since myomesin antibodies

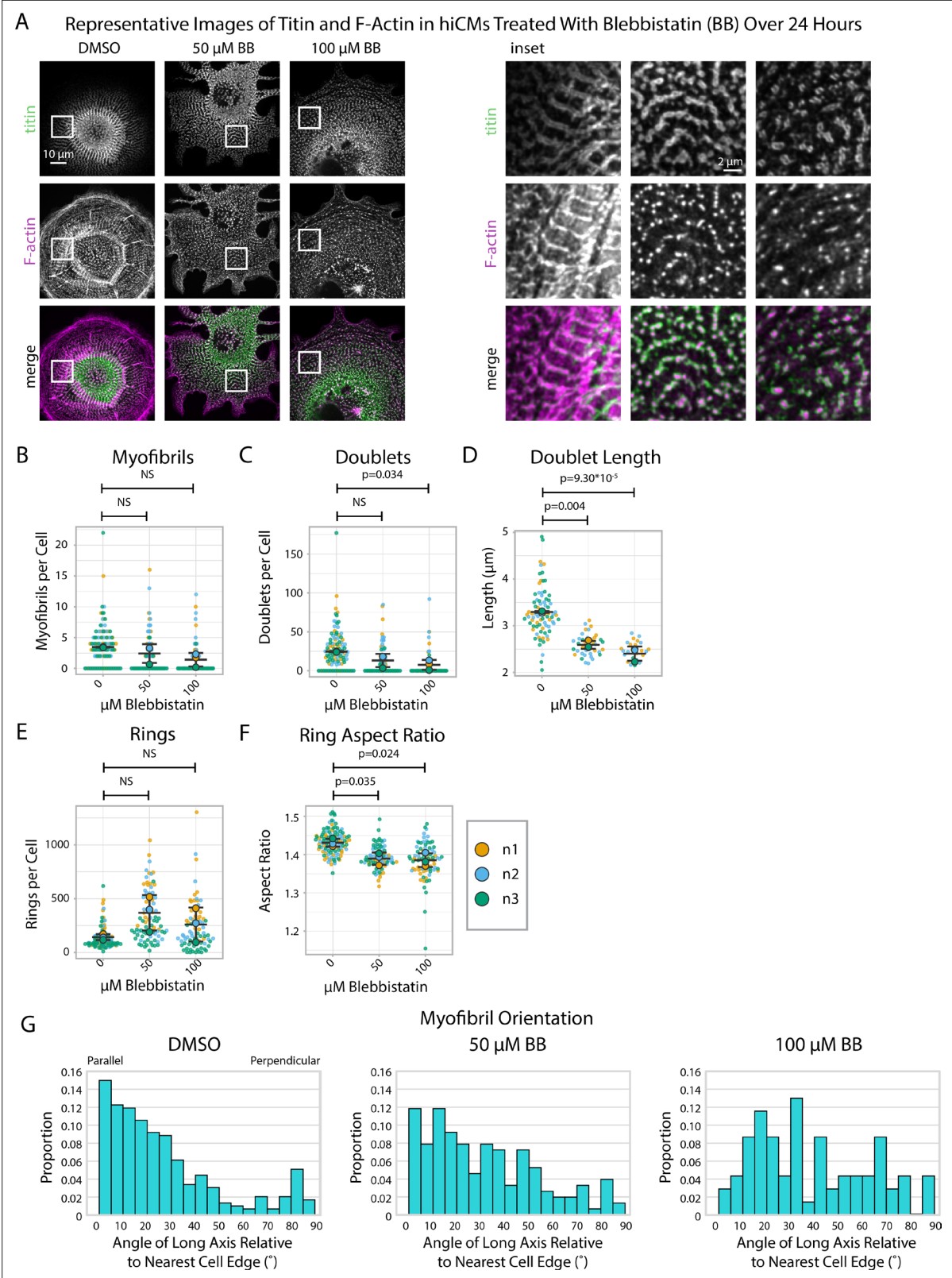

**Figure 5.** Blebbistatin affects myofibril orientation and the morphology of titin structures. (**A**) Representative images of titin and F-actin in human induced pluripotent stem cell-derived cardiac myocytes (hiCMs) treated with DMSO, 50 µM Blebbistatin, and 100 µM Blebbistatin. (**B**) Myofibrils per cell in hiCMs (N = 3 biological replicates, 107 DMSO cells, 95 50 µM Blebbistatin cells, and 84 100 µM Blebbistatin cells). (**C**) Doublets per cell in hiCMs from (**B**). (**D**) Doublet length per cell in hiCMs from (**B**) (N = 58 DMSO cells, 32 50 µM Blebbistatin cells, and 21 100 µM Blebbistatin cells). (**E**) Rings per

*Figure 5 continued on next page*

*Figure 5 continued*

cell in hiCMs from (**B**). (**F**) Average ring aspect ratio per cell in hiCMs from (**B**) (N = 105 DMSO control cells, 95 50 μM Blebbistatin cells, and 78 100 μM Blebbistatin cells). (**G**) Myofibril orientation N = 292 DMSO control myofibrils, 151 50 μM Blebbistatin myofibrils, and 69 100 μM Blebbistatin myofibrils. Full sarcApp outputs can be found in *Figure 5—figure supplement 1*.

The online version of this article includes the following figure supplement(s) for figure 5:

**Figure supplement 1.** Titin quantification and organization in Blebbistatin-treated human induced pluripotent stem cell-derived cardiac myocytes (hiCMs) 24 hr post-plating.

do not stain the edge (see *Figure 2M–O*). To validate sarcApp as a tool to measure M-Lines, we stained myomesin in hiCMs exposed to Blebbistatin. Representative images in *Figure 6E* show that control hiCMs exhibit elongated M-Lines while Blebbistatin-treated M-Lines appear truncated and punctate-like. sarcApp quantification matched observations made by eye, detecting fewer myofibrils per cell (*Figure 6F*) and M-Lines per cell (*Figure 6G*) in Blebbistatin-treated hiCMs, as well as short-ened M-Line length (*Figure 6H*). Co-staining with actin enabled sarcApp quantification of myofibril orientation, revealing that myofibril M-Lines are disorganized relative to the cell edge (*Figure 6I*, see *Figure 2M–O*) akin to the Blebbistatin-induced phenotype seen in Z-Lines (*Figure 3H*).

## Neither α nor β cardiac myosin II alone is required for Z-Line assembly

The sarcomere A-band is the region of thick and thin filament overlap (*Figure 7A*). The thick filament contains stacks of muscle myosins that produce the forces of contraction (*Henderson et al., 2017*). Previous studies probing the role of muscle myosins in sarcomere assembly have produced conflicting results (*Chopra et al., 2018*; *Yang et al., 2018*). With sarcApp capable of measuring both M-Lines and Z-Lines, we were well-poised to ask if and/or how muscle myosins influence the specific assembly of both M-Lines and Z-Lines in hiCMs. Cardiac sarcomeres contain both α myosin II (*MYH6*) and β myosin II (*MYH7*) (*Henderson et al., 2017*; *Chopra et al., 2018*; *Lyons et al., 1990*); we exposed hiCMs to one of two unique sequences targeting either *MYH-* or *MYH7* alongside a non-targeting siRNA (control) then replated, fixed, and stained hiCMs with antibodies to either α-actinin-2 (Z-Lines), titin (Z-Lines), or myomesin (M-Lines) (*Figure 7B–C and H–I*).

After 24 hr, we observed by eye that both the Z-Lines and M-Lines of hiCMs depleted of α myosin II (*MYH6*) appeared to largely resemble those of controls (*Figure 7C*). sarcApp-dependent quan-tification of each stain determined that α myosin II knockdown did not alter length of α-actinin-2 Z-lines, titin doublets, or M-lines, though one of the two siRNAs did result in increased numbers of Z-lines (*Figure 7D–G*). As with α myosin II, stains of β myosin II-depleted hiCMs (*MYH7*) also appeared to resemble those of controls by eye (*Figure 7H and I*); however, sarcApp-based quanti-fication revealed that hiCMs depleted of β myosin II assembled more abundant, but shorter Z-lines than controls (*Figure 7J and K*). Further, while sarcApp detected no change in titin doublet lengths (*Figure 7L*), M-lines were detected as being significantly shorter than controls following β myosin II knockdown (*Figure 7M*). Thus, we conclude that neither α nor β myosin II is strictly required for Z- or M-line assembly in hiCMs but that β myosin II likely influences Z-line concatenation. This is not entirely surprising, as Geach and colleagues showed similar results in *Xenopus* embryos (*Geach et al., 2015*).

## Myomesin is required for A-band but not Z-Line assembly

Given our data suggesting that Z-Line and M-Line assembly could be uncoupled (*Figure 7*), we next asked if directly disrupting M-Line assembly impacts Z-Line assembly. We disrupted M-Line assembly by exposing hiCMs to one of two types of myomesin-targeting siRNAs (gene name: *MYOM*) along-side a non-targeting siRNA (control) then replated, fixed, and stained hiCMs with antibodies to either α-actinin-2 or titin (*Figure 8A and B*).

After 24 hr, we observed by eye that both the α-actinin-2 and titin stains of myomesin-depleted hiCMs resemble those of controls (*Figure 8A*). sarcApp-dependent quantification of α-actinin-2 detected no difference in Z-Lines per cell in myomesin-depleted hiCMs compared to controls (*Figure 8C*), suggesting Z-Lines assemble independently of myomesin; however, sarcApp did detect a decrease in Z-Line length for one myomesin-targeting siRNA (*Figure 8D*, *Figure 8—figure supple-ment 1*). Meanwhile sarcApp-dependent quantification of titin detected no significant change in the number of titin doublets or doublet lengths (*Figure 8E and F*). Interestingly, upon myomesin

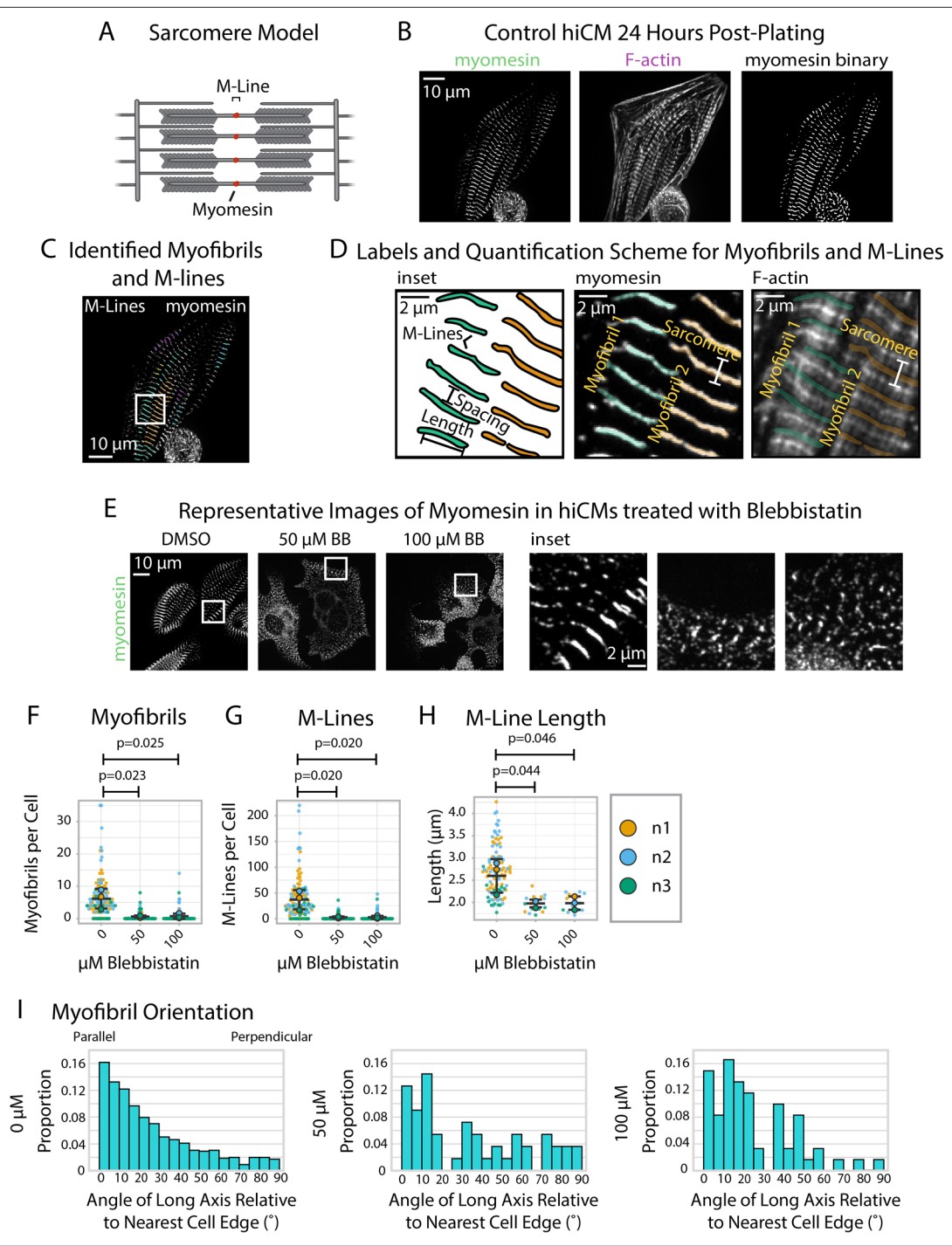

**Figure 6.** sarcApp uses myomesin binaries to identify myofibrils and M-Lines in human induced pluripotent stem cell-derived cardiac myocytes (hiCMs). (**A**) Schematic showing myomesin localization at the M-Line. (**B**) Representative image of myomesin and F-actin in a hiCM. The myomesin binary was predicted using yoU-net as described in the Supplemental Methods. (**C**) Myofibrils and M-Lines identified by sarcApp. Each line denotes an M-Line, and each color represents a myofibril. (**D**) Quantification scheme for myofibrils and M-Lines. Details can be found in *Figure 2—figure supplement 1*. (**E**) Representative images of myomesin and F-actin in hiCMs treated with DMSO, 50 µM Blebbistatin, and 100 µM Blebbistatin. (**F**) Myofibrils per cell in hiCMs (N = 3 biological replicates; 112 DMSO cells, 90 50 µM Blebbistatin cells, and 89 100 µM Blebbistatin cells). (**G**) M-Lines per cell in hiCMs from (**F**). (**H**) Average M-Line length per cell in hiCMs from (**F**) (N = 97 DMSO control cells, 16 50 µM Blebbistatin cells, and 13 100 µM Blebbistatin cells). (**I**) Myofibril orientation (N = 752 DMSO control myofibrils, 49 50 µM Blebbistatin myofibrils, and 60 100 µM Blebbistatin myofibrils). Full sarcApp outputs can be found in *Figure 6—figure supplement 1*.

The online version of this article includes the following figure supplement(s) for figure 6:

*Figure 6 continued on next page*

*Figure 6 continued*

**Figure supplement 1.** Myomesin quantification and organization in Blebbistatin-treated human induced pluripotent stem cell-derived cardiac myocyte (hiCMs) 24 hr post-plating.

knockdown, the number of titin precursor rings significantly increase, warranting future investigation (*Figure 8G*).

We next wanted to assess the influence of myomesin knockdown on M-Line/A-band assembly. Because we were unable to visualize M-Lines in myomesin-depleted hiCMs (data not shown), we stained the A-band using antibodies to the β myosin II motor domain, observing that A-bands in myomesin-depleted hiCMs appear to be less organized (*Figure 8H*) with shorter β myosin II stacks (*Figure 8I*). These data suggest that myomesin regulates A-band/M-Line assembly and myofibril maturation, but not Z-Line assembly. Altogether, these data in tandem with data from *Figure 7* suggest that the assembly of the sarcomere A-band/M-Line and Z-Line are decoupled and/or regulated by mechanisms that are at least partially exclusive to one another.

## Discussion

Here, we introduce sarcApp and yoU-Net, two open-source software packages to binarize microscope images and quantify sarcomere components. We first demonstrate deep learning-based image binarization of three distinct sarcomeric proteins – α-actinin-2, titin, and myomesin – using models trained by a framework we developed called yoU-Net, based off the ubiquitously used U-Net (*Ronneberger and Brox, 2015*). Each trained model is provided and will function as demonstrated herein based on the use of the antibodies as stated in the 'Methods.' yoU-Net is designed and equipped to be functionally versatile, which we hope other researchers can harness to generate additional trained models of proteins not studied here.

The high fidelity of yoU-Net-generated binaries allowed us to quantify several geometric features of the sarcomere in real space as opposed to frequency space. Automatic quantification of sarcomere/myofibril structure with sarcApp offers multiple advantages over traditional, manual quantification including speed, objectivity, and consistency. sarcApp is capable of quantifying 33 measurements per cell and 24 measurements per myofibril at a rate that is orders of magnitude faster than manual quantification, which besides being slower also requires complete attention from the user. sarcApp also removes the potential for user bias/error from quantification and can detect quantitative differences in cells that are not readily apparent by eye (*Figures 7 and 8*) – we expect this will drive discovery of novel mechanisms in cells that would have remained otherwise undetectable. sarcApp as an automatic tool must operate within a rigid quantitative framework and functions optimally when structures o -interest have shapes that are consistent and definable by simple geometric measurements; however, to increase usability we have enabled users to self-define parameters using an easy-to-navigate GUI.

Other than basic structural readouts for sarcomere/myofibril precursors (Z-Bodies, titin rings, etc.) and sarcomeres/myofibrils (Z-Lines, titin doublets, M-Lines, etc.), sarcApp is also equipped to proxy sarcomere/myofibril organization. Myofibril organization correlates with function in vivo and poor organization (disarray) is often cited as causative for myocardial disease (*Eschenhagen et al., 2015*; *Harvey and Leinwand, 2011*; *Olsson et al., 2004*). sarcApp reports organization using a novel metric reported here known as myofibril orientation (*Figure 2M–O*). Orientation is reported relative to the cell membrane by comparing the angle of the myofibril long axis to the nearest cell edge; such a metric was chosen because we routinely observe hiCMs assemble myofibrils with a long axis that is approximately parallel (i.e., 0°) to the cell edge in vitro and also because tissue sections of myocytes in vivo show myofibrillar arrays parallel to the long axis of the membrane (*Fenix et al., 2018*; *Taneja et al., 2020*). We generated this measure of orientation to be more transparent and informative than what is currently available in the hopes of reaching a broader audience (*Chopra et al., 2018*; *Hinson et al., 2015*).

Previous work from our lab and others has included as many cells as reasonably possible in analyses (*Fenix et al., 2018*; *Taneja et al., 2020*), requiring technical expertise in (at minimum) plating cells, sample preparation, microscopy, and image analysis. While microscopy has historically been the rate-limiting step in such analyses, modern microscopes can now rapidly image hundreds to thousands of cells at high resolution. The true bottleneck towards understanding sarcomere assembly now lies for

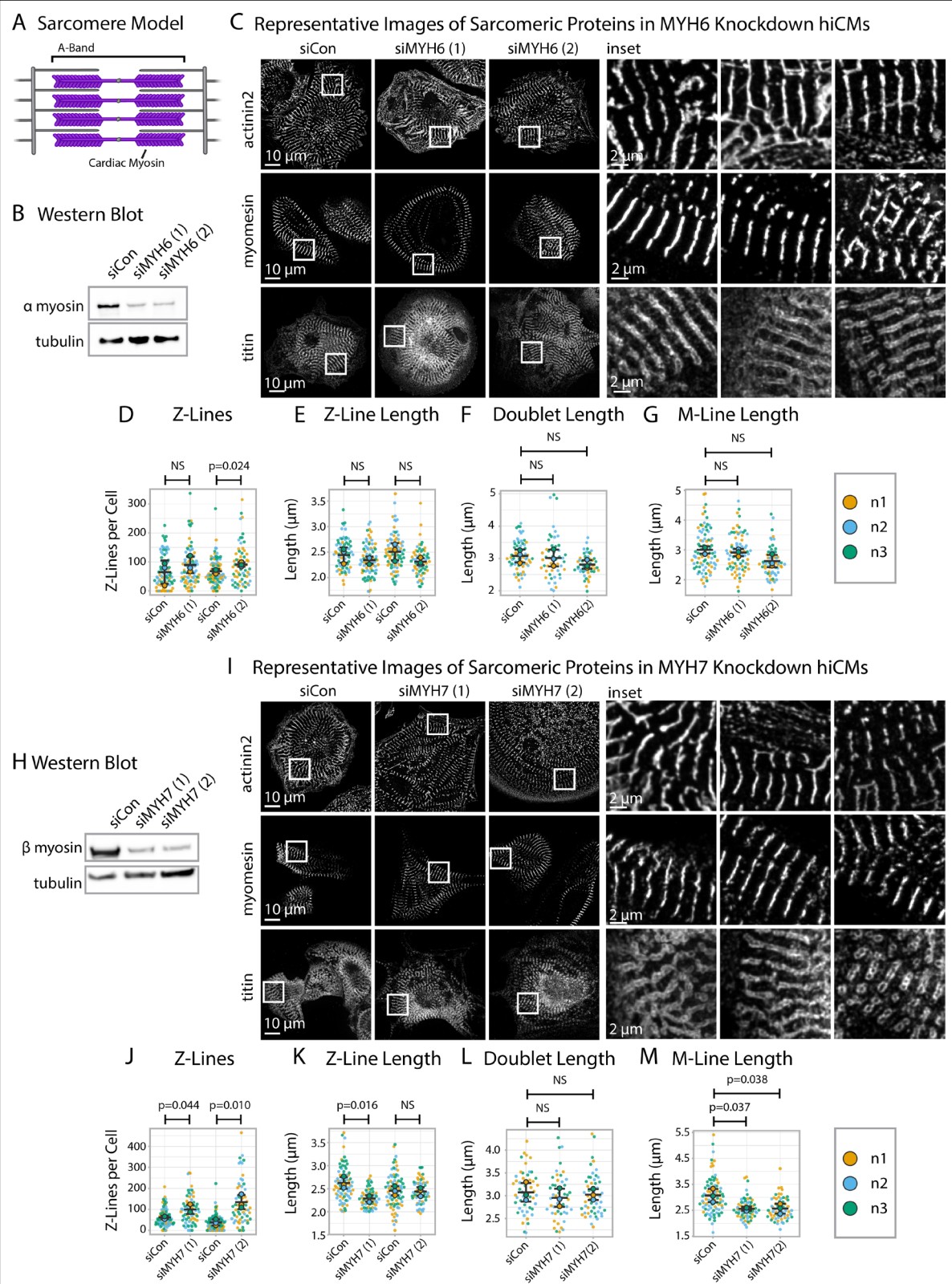

**Figure 7.** Knockdown of α or β cardiac myosin II reduces but does not eliminate sarcomeres. (**A**) Schematic showing cardiac myosin localization in a sarcomere. (**B**) Representative western blot showing α cardiac myosin (MYH6) knockdown in hiCMs. (**C**) Representative images of α-actinin-2, titin, and myomesin in siControl human induced pluripotent stem cell-derived cardiac myocytes (hiCMs) and α cardiac myosin (MYH6) knockdown hiCMs. (**D**) Number of Z-Lines per cell in hiCMs in two independent groups of siControl (scramble)-treated hiCMs and two separate MYH6 siRNA-treated hiCMs

*Figure 7 continued on next page*

*Figure 7 continued*

(sequences 1 and 2). N = 3 biological replicates, 81 siCon cells and 78 siMYH6 (1) cells, and 88 siCon cells and 63 siMYH6 (2) cells. (**E**) Average Z-Line length per cell in hiCMs from (**D**). N = 68 siCon cells and 75 siMYH6 (1) cells, and 83 siCon cells and 62 siMYH6 (2) cells. (**F**) Average doublet length per cell in hiCMs in siCon (scramble)-treated hiCMs and two MYH6 siRNA sequences (1 and 2). 71 siCon cells, 49 siMYH6 (1) cells, and 50 siMYH6 (2) cells. (**G**) Average M-Line length per hiCMs. 85 siCon cells, 78 siMYH6 (1) cells, and 64 siMYH6 (2) cells. (**H**) Representative western blot showing β cardiac myosin (MYH7) knockdown in hiCMs. (**I**) Representative images of α-actinin-2, titin, and myomesin in β cardiac myosin (MYH7) knockdown hiCMs. (**J**) Number of Z-Lines per cell in hiCMs in two independent groups of siCon (scramble)-treated hiCMs and two separate MYH7 siRNA-treated hiCMs (pools 1 and 2). N = 86 siCon cells and 66 siMYH7 (1) cells, and 97 siCon cells and 62 siMYH7 (2) cells. (**K**) Average Z-Line length per cell in hiCMs from (**J**). 81 siCon cells and 63 siMYH7 (1) cells, and 81 siCon cells and 59 siMYH7 (2) cells. (**L**) Average doublet length per hiCM. N = 3 biological replicates, 94 siCon cells, 72 siMYH7 (1) cells, and 66 siMYH7 (2) cells. (**M**) Average M-Line length per cell in hiCMs. N = 87 siCon cells, 65 siMYH7 (1) cells, and 62 siMYH7 (2) cells. Full sarcApp outputs and western blots can be found in *Figure 7—figure supplements 1–7*.

The online version of this article includes the following source data and figure supplement(s) for figure 7:

**Source data 1.** Original 16-bit image file for anti-MYH6 western blot.

**Source data 2.** Original 16-bit image file for anti-tubulin western blot.

**Source data 3.** Original 16-bit image file for anti-MYH7 western blot.

**Source data 4.** Original 16-bit image file for anti-tubulin western blot.

**Figure supplement 1.** α-actinin-2 quantification and organization in MYH6 knockdown human induced pluripotent stem cell-derived cardiac myocytes (hiCMs).

**Figure supplement 2.** Titin quantification and organization in MYH6 knockdown human induced pluripotent stem cell-derived cardiac myocytes (hiCMs).

**Figure supplement 3.** Myomesin quantification and organization in MYH6 knockdown cells.

**Figure supplement 4.** α-actinin-2 quantification and organization in MYH7 knockdown human induced pluripotent stem cell-derived cardiac myocytes (hiCMs).

**Figure supplement 5.** Titin quantification and organization in MYH7 knockdown human induced pluripotent stem cell-derived cardiac myocytes (hiCMs).

**Figure supplement 6.** Myomesin quantification and organization in MYH7 knockdown cells.

**Figure supplement 7.** MYH6 and MYH7 knockdown western blots.

our lab in image analysis/quantification. We summed the quantified cell totals of two recent landmark papers in the field and found the reported totals to be 385 (*Chopra et al., 2018*) and 785 (*Fenix et al., 2018*). sarcApp has allowed us to present in this article multidimensional outputs from 3452 single cells with outputs that we feel represent the data clearly at both basic glance and upon close inspection. Much like yoU-net, we have encoded sarcApp into a GUI equipped with a user manual that details the steps one needs to use sarcApp, irrespective of coding expertise.

We validated sarcApp using the pan-myosin II inhibitor Blebbistatin (*Limouze et al., 2004*). Blebbistatin exposure results in predictable, dose-dependent inhibition of sarcomere assembly (*Chopra et al., 2018*; *Liu et al., 2016*). sarcApp-generated measurements of α-actinin-2 in the presence or absence of Blebbistatin were consistent with observations made by previous groups (*Chopra et al., 2018*; *Liu et al., 2016*; *Figure 3*). Meanwhile, several other outputs related to titin and myomesin first reported here by sarcApp suggest that periodic, pseudomyofibrillar structures characteristic to both the Z-Line and M-Line can at least partially self-assemble in the absence of myosin II motor-based contractility (*Figures 5 and 6*). These data indicate other mechanisms beyond myosin II-based motor-based contractility may independently contribute to the assembly and ordered arrangement of myofibrils in hiCMs. Specifically with titin, we also observe the phenomenon that Blebbistatin disrupts the higher-order organization of titin-based structures with respect to the cell membrane but does not affect titin periodicity within the structure. These data suggest that unique mechanistic tiers of organization exist within the cell at the sub- and super-structural level that become uncoupled upon relieving the cell of myosin II-dependent contractility.

We find that neither α nor β cardiac myosin II knockdown had any measurable impact on the assembly of Z-Lines (*Figure 7*). Attempts to determine if Z-Lines assemble in the simultaneous absence of both myosins were unsuccessful in our hands. Such an experiment, in addition to the Blebbistatin experiments reported herein and elsewhere (*Chopra et al., 2018*; *Liu et al., 2016*), could resolve whether myosin II-dependent contractility alone is sufficient to explain the role of myosin II in Z-Line assembly. Meanwhile, β cardiac myosin II knockdown did result in hiCMs with shorter M-Lines,

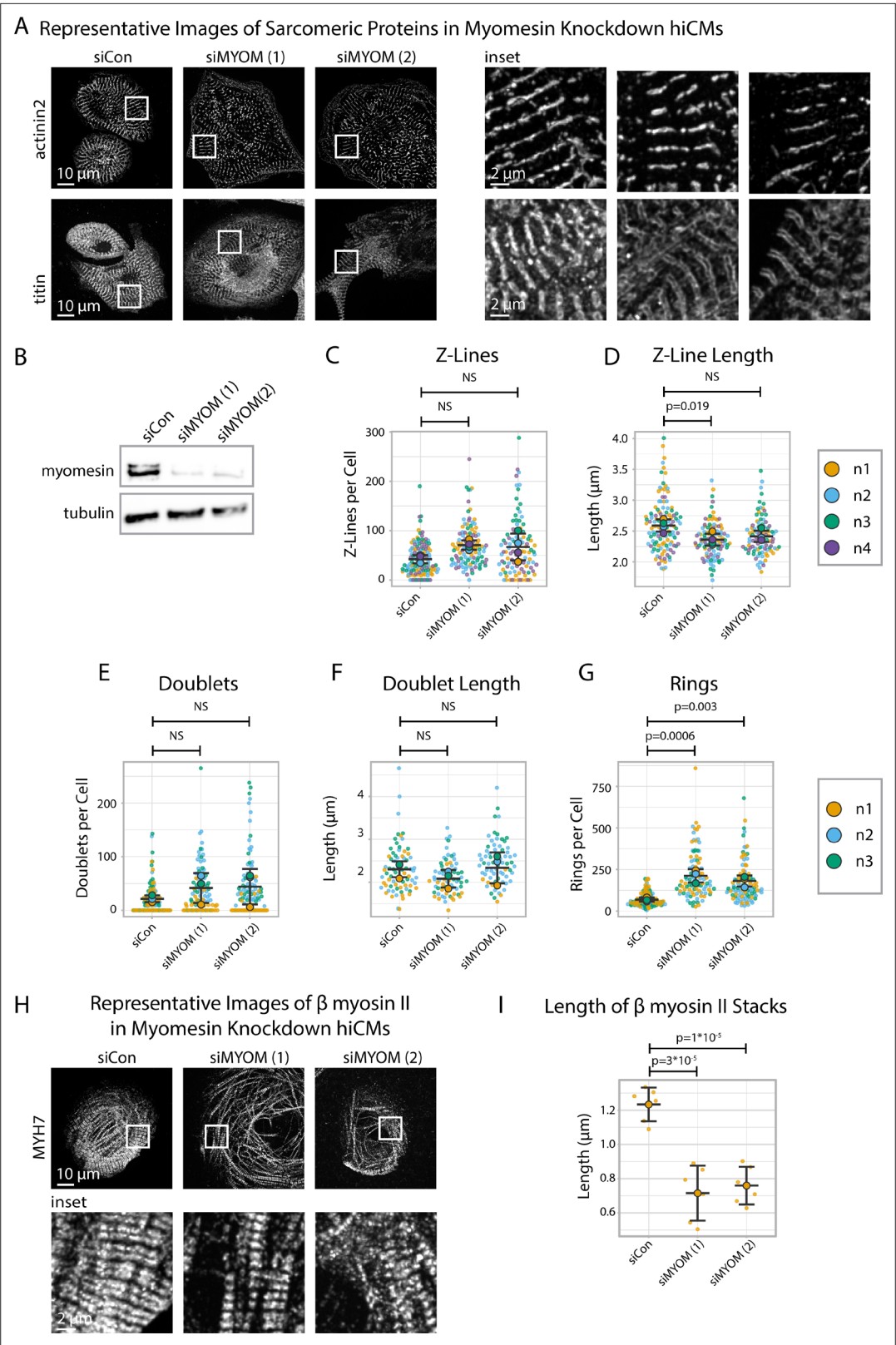

**Figure 8.** Myomesin knockdown alters titin and cardiac myosin II localization, but not α-actinin-2. (**A**) Representative images of sarcomeric proteins α-actinin-2 and titin in myomesin (MYOM) knockdown human induced pluripotent stem cell-derived cardiac myocytes (hiCMs). (**B**) Representative western blot and quantification showing MYOM knockdown in hiCMs, N = 3 biological replicates. (**C**) Number of Z-Lines per cell in siCon

*Figure 8 continued on next page*

*Figure 8 continued*

(scramble)-treated hiCMs and two separate MYOM siRNA-treated hiCMs (sequences 1 and 2). N = 3 biological replicates, 132 siCon cells, 105 siMYOM (1) cells, and 103 siMYOM (2) cells. (**D**) Average Z-Line length per cell in hiCMs from (**C**): N = 4 biological replicates, 117 siCon cells, 104 siMYOM (1) cells, and 92 siMYOM (2) cells (only cells with myofibrils were quantified for **D**). (**E**) Titin doublets per cell in hiCMs using experimental treatments in (**C**). N = 3 biological replicates, 117 siCon cells, 90 siMYOM (1) cells, and 100 siMYOM (2) cells. (**F**) Average doublet length per cell in hiCMs from (**E**): N = 3 biological replicates, 74 siCon cells, 68 siMYOM (1) cells, and 70 siMYOM (2) cells (only cells with myofibrils were quantified for **F**). (**G**) Rings per cell in hiCMs from (**E**). (**H**) Representative images of MYH7 and F-actin in siCon, siMYOM (1), and siMYOM (2) hiCMs. (**I**) Length of β cardiac myosin stacks in siCon, siMYOM (1), and siMYOM (2) hiCMs. Full sarApp outputs and western blot can be found in ***Figure 8— figure supplements 1–3***.

The online version of this article includes the following source data and figure supplement(s) for figure 8:

**Source data 1.** Original 16-bit image file for anti-MYH6 western blot.

**Source data 2.** Original 16-bit image file for anti-tubulin western blot.

**Figure supplement 1.** α-actinin-2 quantification and organization in myomesin knockdown human induced pluripotent stem cell-derived cardiac myocytes (hiCMs).

**Figure supplement 2.** Titin quantification and organization in myomesin knockdown human induced pluripotent stem cell-derived cardiac myocytes (hiCMs).

**Figure supplement 3.** MYOM knockdown western blots.

---

indicating that either contractility or scaffolding by β, but not α myosin II, is required for M-Line maturation (***Figure 7G and M***). Knockdown of myomesin similarly did not impact Z-Line assembly but did result in shortening of the thick filament A-band (***Figure 8H and I***). Our ability to perturb the M-Line/A-band but not the Z-Line suggests that Z- and M-Line assembly are regulated at least in part by distinct mechanistic axes, as previously hypothesized (***Rudy et al., 2001***; ***Schafer et al., 1995***; ***Beall et al., 1989***). It should be noted here that solely using Z-Lines as a proxy for sarcomere status would have led us to errantly conclude that β cardiac myosin II and myomesin both play no or at least minor roles in sarcomere assembly. Future studies examining assembly should therefore consider Z-Line assembly to be distinct from, or at least not wholly representative of, assembly of other sarcomere components.

Here we used exclusively fixed-cell immunofluorescence for high-throughput quantification of sarcomere assembly. We eliminated as much bias as possible by selecting cells for quantification solely using a nuclear DAPI stain and are therefore confident that data presented herein fully captures the true biological variability present within whole hiCM cultures. A future goal of the lab is to further adapt sarcApp for live cell quantification. Tracking specific structures across time would be a necessity for a 'live' version of sarcApp, a hurdle the lab is currently working to overcome. We also designed this initial version of sarcApp to quantify in 2D rather than in 3D space. As we have published previously, sarcomeres/myofibrils assemble on the dorsal (top) surface of an hiCM plated on glass for up to 48 hr after plating, and we consider them within that timeframe to exist largely within a two-dimensional plane (***Fenix et al., 2018***; ***Taneja et al., 2020***). Breaking into three dimensions will require more complex mathematics and produce more convoluted outputs, which could limit adoption by other researchers, but is a long-term goal of the lab nonetheless.

In summary, sarcApp and yoU-Net provide frameworks for automatic binarization, segmentation, and quantitative analysis of sarcomeric proteins in striated muscle cells. We anticipate these software packages will be useful for lower-throughput single-cell analyses of sarcomere structure and/ or assembly as well as for high-throughput screens to analyze several hundreds to thousands of cells at a time. In this way, we hope to facilitate the entry of more researchers into the field of cardiac cell biology and continue to introduce novel quantification metrics and additional sarcomere proteins in the future through continued maintenance and management of sarcApp as an open-source community project.

## Methods

SarcApp and yoU-net can be found at the following address:

https://github.com/abbieneininger/sarcApp; (copy archived at ***Abbieneininger, 2023***).

**Key resources table**

| Reagent type (species) or resource | Designation | Source or reference | Identifiers | Additional information |
|---|---|---|---|---|
| Antibody | Anti-alpha actinin 2 (mouse monoclonal) | Sigma | A7811 | IF, 1:200 |
| Antibody | Anti-Titin (mouse monoclonal) | DHSB | 9D10 | IF, 1:2 |
| Antibody | Mouse anti-Myomesin | DHSB | MYOM | IF, 1:2 |
| Antibody | Mouse anti-MYH | DHSB | A4.591 | IF, 1:2 |
| Antibody | Rabbit anti-MYH6 | ProteinTech | 22281-1-AP | WB, 1:500 |
| Antibody | Rabbit anti-MYH7 | ProteinTech | 22280-1-AP | WB, 1:500 |
| Antibody | Goat anti-mouse 488 | Life Technologies | A11001 | IF, 1:100 |
| Antibody | Goat anti-rabbit 488 | Life Technologies | A11034 | IF, 1:100 |
| Antibody | Goat anti-mouse 568 | Life Technologies | A11004 | IF, 1:100 |
| Antibody | Goat anti-rabbit 568 | Life Technologies | A11036 | IF, 1:100 |
| Antibody | Goat anti-mouse 647 | Life Technologies | A32728 | IF, 1:100 |
| Antibody | Goat anti-rabbit 647 | Life Technologies | A32733 | IF, 1:100 |
| Biological sample (*Bos taurus*) | Bovine serum albumin | RPI | A30075-100 | |
| Chemical compound, drug | Phalloidin Alexa Fluor 488 | Invitrogen | A12379 | |
| Chemical compound, drug | Phalloidin Alexa Fluor 568 | Invitrogen | A12380 | |
| Chemical compound, drug | Phalloidin Alexa Fluor 647 | Invitrogen | A22287 | |
| Chemical compound, drug | PBS, 10×, $Ca^{2+}/Mg^{2+}$ free | Life Technologies | 70011-044 | |
| Chemical compound, drug | PBS, 10×, with $Ca^{2+}/Mg^{2+}$ | Corning | 46-013CM | |
| Chemical compound, drug | PFA, 16% | Electron Microscopy Sciences | 15710 | |
| Chemical compound, drug | 0.5% Trypsin | Life Technologies | 15400-054 | |
| Chemical compound, drug | 0.1% Gelatin | Sigma | ES-006-B | |
| Chemical compound, drug | Dimethyl sulfoxide | Sigma | 276855 | |
| Chemical compound, drug | Vectashield with DAPI | Vector | H-1200 | |
| Chemical compound, drug | Blebbistatin | Sigma | B0560 | |
| Chemical compound, drug | Fibronectin | Corning | 354008 | |
| Chemical compound, drug | Lipofectamine RNAiMAX | Thermo Fisher | LMRNA015 | |
| Chemical compound, drug | TBS, 10× | Corning | 46-012CM | |
| Chemical compound, drug | Tween 20 | Sigma | P7949 | |
| Cell line (*Homo sapiens*) | iCell cardiomyocytes^2 kit | Fujifilm International | CMC-100-012-000.5 | |
| Sequence-based reagent | SMART Pool siRNA against human MYH7 (13–16) | Horizon Discovery | A-011086 | |
| Sequence-based reagent | SMART Pool siRNA against human MYH7 (13, 14) | Horizon Discovery | A-011086-13, A-011086-14 | |
| Sequence-based reagent | siRNA against human MYH6 3' UTR (13) | Horizon Discovery | A-012645-13-0005 | |
| Sequence-based reagent | siRNA against human MYH6 CDS (14) | Horizon Discovery | A-012645-14-0005 | |
| Sequence-based reagent | siRNA against human myomesin 3'UTR (13) | Horizon Discovery | A-006342-13-0005 | |
| Sequence-based reagent | siRNA against human myomesin CDS (16) | Horizon Discovery | A-006342-16-0005 | |
| Software, algorithm | FIJI | NIH | | |

*Continued on next page*

*Continued*

| Reagent type (species) or resource | Designation | Source or reference | Identifiers | Additional information |
|---|---|---|---|---|
| Software, algorithm | sarcApp | This study | | |
| Software, algorithm | yoU-Net | This study | | |

### The supplemental user manual and sarcApp software

Details on sarcApp and yoU-Net (*Neininger-Castro, 2023*) download and usage documentation are given in the supplement and at https://github.com/abbieneininger/sarcApp; (copy archived at *Abbieneininger, 2023*). Image data is available upon request.

### Cell culture and authentication

Human iPSC-derived cardiac myocytes (CMM-100-012-000.5, Cellular Dynamics, Madison, WI) were cultured as per the manufacturer's instructions in proprietary manufacturer-provided cardiac myocyte maintenance medium in polystyrene 96-well cell culture plates. Cells were maintained at 37°C and 5% $CO_2$. For replating hiCMs onto glass substrates, cells were washed two times with 100 µL 1× PBS with no $Ca^{2+}$/$Mg^{2+}$ (PBS*, 70011044, Gibco, Grand Island, NY). PBS* was completely removed from hiCMs and 40 µL 0.1% Trypsin-EDTA with no phenol red (15400054, Gibco) was added to hiCMs and incubated at 37°C for 2 min. Following incubation, the cells were washed three times with trypsin, the plate rotated 180°, and washed another three times. Trypsinization was then quenched by adding 120 µL of culture media and total cell mixture was pipetted into a 1.5 mL Eppendorf tube. Cells were centrifuged at 200 × *g* for 3 min, and the supernatant was aspirated. The cell pellet was resuspended in 200 µL of culture media and plated on 35 mm dishes with a 10 mm glass bottom (D35-10-1.5-N; CellVis, Sunnydale, CA) pre-coated with 10 µg/mL fibronectin (354008, Corning) for 1 hr at 37°C.

### Antibodies

Alexa Fluor 488-phalloidin (A12379), Alexa Fluor 568-phalloidin (A12380), and Alexa Fluor 647-phalloidin (A22287) were purchased from Invitrogen. Alexa Fluor 488-goat anti-mouse (A11029), Alexa Fluor 488-goat anti-rabbit (A11034), Alexa Fluor 568-goat-anti-rabbit (A11011), Alexa Fluor 568-goat anti-mouse (A11004), Alexa Fluor 647-goat-anti-mouse (A32728), and Alexa Fluor 647-goat-anti-rabbit (A32733) (1:100) antibodies were purchased from Life Technologies (Grand Island, NY).

The Titin (9D10) antibody, Myomesin (MYOM) antibody, and MYH7 (A4.591) (all 1:2) antibody were purchased from the Developmental Studies Hybridoma Bank (University of Iowa). Mouse anti-α-actinin-2 (1:200, A7811) was purchased from Sigma-Aldrich. Rabbit anti-MYH6 and MYH7 for western blotting (1:500) were purchased from ProteinTech (22281-1-AP, 22280-1-AP).

### Chemicals

Blebbistatin was purchased from Sigma (B0560) and reconstituted to 10 mM in DMSO (Sigma 276855).

### Fixation and Immunostaining

Cells were fixed with 4% paraformaldehyde (PFA) in PBS at room temperature for 20 min and then permeabilized for 5 min with 1% Triton X-100/4% PFA in PBS. For actin visualization, phalloidin Alexa 488 or Alexa 568 in 1× PBS (15 µL of stock phalloidin per 200 µL of PBS) was used for 2 hr at room temperature. For immunofluorescence experiments, cells were blocked in 5% bovine serum albumin (BSA) in PBS for 20 min, followed by antibody incubations.

For visualizing titin and myomesin, a live-cell extraction was performed to remove cytoplasmic background (*Svitkina, 2009*). A cytoskeleton-stabilizing live-cell extraction buffer was made fresh containing 2 mL of stock solution (500 mM 1,4-piperazinediethanesulfonic acid, 25 mM ethylene glycol tetra acetic acid, 25 mM $MgCl_2$), 4 mL of 10% polyoxyethylene glycol (PEG; 35,000 molecular weight), 4 ml $H_2O$, and 100 µL of Triton X-100, 10 µM paclitaxel, and 10 µM phalloidin. Cells were treated with this extraction buffer for 1 min, followed by a 1 min wash with wash buffer (extraction

buffer without PEG or Triton X-100). Cells were then fixed with 4% PFA for 20 min, followed by antibody labeling. VectaShield with DAPI (H-1200, Vector Laboratories Inc, Burlingame, CA) was used for mounting.

## Protein knockdown

Knockdowns for MYOM were performed using single siRNAs from GE Dharmacon: one to the UTR of MYOM and one to the coding sequence (CDS). Knockdowns for MYH6 were performed using one of two siRNAs for MYH6: one to the UTR and one to the CDS purchased from GE Dharmacon. Knockdowns for MYH7 were performed using one of two siRNA pools: one of two siRNAs and one of four. Experiments were performed in 96-well culture plates, using the Lipofectamine RNAiMAX reagent and instructions provided by the manufacturer (Thermo Fisher, LMRNA015). Following knockdown, cells were replated onto glass substrates for 24 hr and fixed for immunofluorescence or lysed for western blotting.

## Western blotting

Gel samples were prepared by mixing cell lysates with LDS sample buffer (Life Technologies, NP0007) and sample reducing buffer (Life Technologies, NP0009) and boiled at 95°C for 5 min. Samples were resolved on Bolt 4–12% gradient Bis-Tris gels (Life Technologies, NW04120BOX). Protein bands were blotted onto a nylon membrane (Millipore). Blots were blocked using 5% nonfat dry milk (NFDM, Research Products International, Mt. Prospect, IL, M17200) in Tris-buffered saline with Tween-20 (TBST, 10× TBS from Corning 46-012CM, Tween-20 from Sigma P7949). Antibody incubations were also performed in 5% NFDM in TBST. Blots were developed using the Immobilon Chemiluminescence Kit (Millipore, WBKLS0500).

## Fluorescence microscopy

Imaging was performed using either Spinning Disk confocal microscopy or instant Structured Illumination Microscopy (iSIM). The spinning disk images were taken on a Nikon Spinning Disk confocal microscope equipped with Apo TIRF Oil 100 × 1.49 NA objective and a Photometrics Prime 95B cMOS monochrome camera, provided by the Nikon Center of Excellence, Vanderbilt University. Images were deconvolved post-acquisition using the FIJI Microvolution software plugin (Microvolution, Cupertino, CA). iSIM imaging was performed with a Visitech iSIM using a Nikon SR HP Apo TIRF 100× oil immersion objective (model number MRD01997) at 1× zoom with NA = 1.49. Images were captured using a Hamamatsu ORCA-Fusion Digital CMOS camera (model C14440-20UP) with a 0.1 µm axial step size. Images were deconvolved using Microvolution software (Cupertino, CA) installed in FIJI (Fiji Is Just ImageJ) over 20 iterations.

## Platinum replica transmission EM of live-cell extracted cells

Adherent plasma membranes from cultured cardiomyocytes grown on glass coverslips were detergent extracted. Cells were treated with extraction buffer (2 mL stock buffer [5× Stock Buffer: 500 mM 1,4-piperazinediethanesulfonic acid, 25 mM ethylene glycol tetraacetic acid, 25 mM MgCl$_2$, pH'd and kept at 4°C], 4 mL 10% PEG [35,000 MW], 4 mL milliQ H$_2$O, 100 uL of TritonX-100, 10 uM nocodazole, and 10 uM phalloidin) for 30 min, followed by a 1 min wash with wash buffer (2 mL stock buffer, 8 mL milliQ H$_2$O, 10 uM nocodazole, 10 uM phalloidin), followed by fixation (2% PFA, 2% glutaraldehyde) for 20 min. Extracted cells were further sequentially treated with 0.5% OsO$_4$, 1% tannic acid, and 1% uranyl acetate before graded ethanol dehydration and hexamethyldisilazane (HMDS) substitution (LFG Distribution, France). Dried samples were then rotary shadowed with 2 nm of platinum (sputtering) and 4–6 nm of carbon (carbon thread evaporation) using an ACE600 metal coater (Leica Microsystems, Germany). The resultant platinum replica was floated off the glass with hydrofluoric acid (5%), washed several times on distilled water, and picked up on 200 mesh formvar/carbon-coated EM grids. The grids were mounted in a eucentric side-entry goniometer stage of a transmission electron microscope operated at 120 kV (JEOL, Japan), and images were recorded with a Xarosa digital camera (EM-SIS, Germany). Images were processed in Adobe Photoshop to adjust brightness and contrast and presented in inverted contrast.

## Statistical analyses

Analyses comparing three groups (one control and two treatment groups: *Figures 3C–G, 5B–F, 6F–H, 7D–H, K–O, and 8C–G*, *Figure 3—figure supplement 1A–L*, *Figure 3—figure supplement 2A–L*, *Figure 3—figure supplement 3A–L*, *Figure 5—figure supplement 1A–N*, *Figure 6—figure supplement 1A–G*, *Figure 7—figure supplement 2A–N*, *Figure 7—figure supplement 3A–G*, *Figure 7—figure supplement 6A–G*, *Figure 8—figure supplement 1A–L*, *Figure 8—figure supplement 2A–N*) were calculated using a one-way ANOVA. If significant, a post-hoc Tukey test (*Figures 3C–F, 5D, 6F–H, 7E, K–M, and 8D–G*, *Figure 3—figure supplement 2C, D, F, G, K*, *Figure 3—figure supplement 3A–D*, *Figure 5—figure supplement 1C, D, K, M*, *Figure 6—figure supplement 1A–D*, *Figure 7—figure supplement 2N*, *Figure 7—figure supplement 3F, G*, *Figure 7—figure supplement 5E, F, N*, *Figure 7—figure supplement 6D, F, G*, *Figure 8—figure supplement 1A, D, K*, *Figure 8—figure supplement 2F, I, L, N.*) was done. Analyses comparing two groups were calculated using a two-tailed unpaired Student's $t$-test (*Figure 8A–L, S11A–L*, *Figure 7—figure supplement 4A-L*). Graphs were made using SuperPlots (*Lord et al., 2020*).

## Acknowledgements

We would like to thank the instructors, TAs, and fellow classmates at the Marine Biosciences Laboratory course Deep Learning for Biological Microscopy, especially Jan Funke (Janelia), Dagmar Kainmueller (MDC Berlin), Zachary Whiddon (University of Louisville), and William Patton (Janelia) for assistance on early versions of yoU-Net. We would also like to thank Bryan Millis at the Vanderbilt Biophotonics Center and Kari Seedle at the Vanderbilt Nikon Center for Excellence and the Vanderbilt Center for Imaging Shared Resources (CISR) for experimental and imaging assistance. We thank Vanderbilt's Program in Developmental Biology, Microtubules and Motors Club and Molecular Biophysics Training Program for project feedback and discussion. This work was supported by Vanderbilt University T32 5T32HD007502-25 to ACN and ZCS, Vanderbilt University T32 5T32GM008320-32 to JBH, American Heart Predoctoral Fellowship 836090 to JBH and 18PRE33960551 to NT, NIH NIGMS R35 GM125028 to DTB, Vanderbilt University R25 5R25GM062549-18 and American Heart Predoctoral Fellowship 1070985 to ZCS, and F31 HL136081 to AMF.

## Additional information

### Funding

| Funder | Grant reference number | Author |
|---|---|---|
| National Institute of General Medical Sciences | R35 GM125028 | Dylan Tyler Burnette |
| Eunice Kennedy Shriver National Institute of Child Health & Human Development | T32 5T32HD007502 | Abigail C Neininger-Castro Zachary C Sanchez |
| National Institute of General Medical Sciences | T32 5T32GM008320 | James B Hayes |
| American Heart Association | 836090 | James B Hayes |
| American Heart Association | 1070985 | Zachary C Sanchez |
| National Heart, Lung, and Blood Institute | F31 HL136081 | Aidan M Fenix |
| American Heart Association | 18PRE33960551 | Nilay Taneja |

The funders had no role in study design, data collection and interpretation, or the decision to submit the work for publication.

## Author contributions
Abigail C Neininger-Castro, Conceptualization, Data curation, Software, Formal analysis, Validation, Investigation, Visualization, Methodology, Writing – original draft, Writing – review and editing; James B Hayes, Funding acquisition, Investigation, Writing – review and editing; Zachary C Sanchez, Funding acquisition, Investigation, Methodology; Nilay Taneja, Aidan M Fenix, Conceptualization, Funding acquisition, Writing – review and editing; Satish Moparthi, Stéphane Vassilopoulos, Investigation, Writing – review and editing; Dylan Tyler Burnette, Conceptualization, Funding acquisition, Methodology, Writing – original draft, Writing – review and editing

## Author ORCIDs
Stéphane Vassilopoulos (iD) http://orcid.org/0000-0003-0172-330X
Dylan Tyler Burnette (iD) http://orcid.org/0000-0002-2571-7038

Reviewer #1 (Public Review): https://doi.org/10.7554/eLife.87065.3.sa1
Reviewer #2 (Public Review): https://doi.org/10.7554/eLife.87065.3.sa2
Reviewer #3 (Public Review): https://doi.org/10.7554/eLife.87065.3.sa3
Author Response https://doi.org/10.7554/eLife.87065.3.sa4

---

# Additional files

## Supplementary files
• MDAR checklist

## Data availability
SarcApp and yoU-Net software can be found at Github. https://github.com/abbieneininger/sarcApp, (copy archived at *Abbieneininger, 2023*).

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
