## [Editor Report · eLife assessment]

This manuscript describes a **useful** tool for quantitative assessment of sarcomere structures in healthy and perturbed cardiomyocytes grown in vitro. The work is **solid**, and the methods, data and analyses broadly support the claims with only minor weaknesses. The tool will be relevant to biologists working on and interested in obtaining quantitative information on sarcomere structure, function and development.

---

## [Referee Report · Reviewer #1 (Public Review)]

This manuscript by Neininger-Castro and colleagues presents a novel automatic image analysis method for assessing sarcomeres, the basic units of myofibrils and validates this tool in a couple of experimental approaches that interfere with sarcomere assembly in iPSC-cardiomyocytes (iPSC-CM).

Automatic quantification of sarcomeres is definitely something that is useful to the field. I am surprised that there is no reference in the manuscript to SarcTrack, published by Toepfer and colleagues in 2019 (PMID 30700234), which has exactly the same purpose. The advantage of the image analysis software presented in the current manuscript appears to me to be that it can cover both mature sarcomeres and nascent sarcomeres in premyofibrils effectively.

---

## [Referee Report · Reviewer #2 (Public Review)]

Neininger-Castro et al report on their original study entitled "Independent regulation of Z-lines and M-lines during sarcomere assembly in cardiac myocytes revealed by the automatic image analysis software sarcApp", In this study, the research team developed two software, yoU-Net and sarcApp, that provide new binarization and sarcomere quantification methods. The authors further utilized human induced pluripotent stem cell-derived cardiomyocytes (hiCMs) as their model to verify their software by staining multiple sarcomeric components with and without the treatment of Blebbistatin, a known myosin II activity inhibitor. With the treatment of different Blebbistatin concentrations, the morphology of sarcomeric proteins was disturbed. These disrupted sarcomeric structures were further quantified using sarcApp and the quantification data supported the phenotype. The authors further investigated the roles of muscle myosins in sarcomere assembly by knocking down MYH6, MYH7, or MYOM in hiCMs. The knockdown of these genes did not affect Z-line assembly yet the knockdown of MYOM affected M-line assembly. The authors demonstrated that different muscle myosins participate in sarcomere assembly in different manners.

---

## [Referee Report · Reviewer #3 (Public Review)]

Neininger-Castro and colleagues developed software tools for the quantification of sarcomeres and sarcomere-precursor features in immunostained human induced pluripotent stem cell-derived cardiac myocytes (hiCMs). In the first part they used a deep-learning- based model called a U-Net to construct and train a network for binarization of immunostained cardiomyocyte images. They also wrote graphical user interface (GUI) software that will assist other labs to use this approach and made it publicly available. They did not compare their approach to existing ones, but example from one image suggests their binarization tool outperforms Otsu thresholding binarization.

In the second part they developed a software tool called sarcApp that classifies sarcomere structures in the binarized image as a Z-Line or Z-Body and assigns each to either a myofibril or to stress fibers. The tools can then automatically count and measure multiple features (33 per cell and 24 per myofibril) and report them on a per-cell, per-myofibril, and per- stress fiber basis.

To test the tools they used Blebbistatin to inhibit sarcomere assembly and showed that the sarcApp tool could capture changes in multiple features such as fewer myofibrils, fewer Z-Lines, decreased myofibril persistence, decreased Z-Line length and altered myofibril orientation in the Blebbistatin treated cells. With some changes the tool was also shown to quantify sarcomeres in titin and myomesin stained cardiomyocytes.

Finally they used sarcApp to quantify the changes in sarcomere assembly after siRNA mediated knockout of MYH7, MYH7, or MYOM. The analysis indicates that neither MYH6 nor MYH7 knockdown perturbed the assembly of Z- or M-lines, and that knockdown of MYOM perturbed the A-band/M-Line but not the Z-Line assembly according to features captured by the sarcApp tool.

Overall the authors developed and made publicly available an excellent software tool that will be very useful for labs that are interested in studying sarcomere assembly. Multiple features that are difficult to measure or count manually can be automatically measured by the software quickly and accurately.

There are however some remaining questions about these tools:

1. The binarization tool which is tailored to sarcomere image binarization appears promising but was not systematically compared with existing approaches. Example from one cell suggests it outperforms Otsu's binarization approach.

2. How robust is the tool? The tool was tested on images from one type of cardiomyocytes (hiCMs) taken from one lab using Nikon Spinning Disk confocal microscope equipped with Apo TIRF Oil 100X 1.49 NA objective or instant Structured Illumination Microscopy (iSIM), using deconvolution (Microvolution software) and in a specific magnification. It remains to be seen whether the tool would be equally effective with images taken with other microscopy systems, with other cardiomyocytes (chick or neonatal rat), with different magnifications, live imaging, etc. The authors state that this approach is also useful in other situations, but the data is not included in this manuscript.

3. The tool was developed for evaluation of sarcomere assembly. The authors show that for this application it can detect the perturbation by Blebbistatin, or knockdown of sarcomeric genes. It remains to be seen if this tool is also useful for assessment of sarcomere structure for other questions beside sarcomere assembly and in other sarcomere pathologies.

---

## [Author Response]

The following is the authors’ response to the original reviews.

**Reviewer #1 (Public Review):**
This manuscript by Neininger-Castro and colleagues presents a novel automatic image analysis method for assessing sarcomeres, the basic units of myofibrils and validates this tool in a couple of experimental approaches that interfere with sarcomere assembly in iPSCcardiomyocytes (iPSC-CM).Automatic quantification of sarcomeres is definitely something that is useful to the field. I am surprised that there is no reference in the manuscript to SarcTrack, published by Toepfer and colleagues in 2019 (PMID 30700234), which has exactly the same purpose. The advantage of the image analysis software presented in the current manuscript appears to me to be that it can cover both mature sarcomeres and nascent sarcomeres in premyofibrils effectively.

We whole-heartedly disagree that SarcTrack has the exact same purpose as sarcApp. sarcApp measures more than the frequency of actinin2 images, and can measure real-space quantifications of actinin, myomesin, and titin, which has not been done before in this way. However, SarcTrack is an interesting method that we hope many researchers find helpful in their research. SarcTrack is a particle tracker that outputs the dimensions of the objects found, but does not distinguish between Z-Lines and other actinin2-positive structures (Z-Bodies, adhesions). It also does not group these structures into higher order structures such as myofibrils and muscle stress fibers.

When going through the manuscript there were a few issues that should be addressed in a revised version of the manuscript:1. I am a bit puzzled that they took 1.4 um length as a cutoff length for a mature A-band in their quantifications, since the consensus in the field for thick filament length seems to be 1.6 um?

We use 1.4 µm as a cutoff length for the length of a Z-Line rather than the A-Band. We believe the reviewer is referring to the width of the A-Band perpendicular to the Z-lines, which is indeed 1.6 µm. However, we are referring to the length of the Z-Lines, which can span anywhere from1.4 µm to up to 10 or more µm. Thank you for allowing us to make the clarification.

1. When doing the knockdown for alpha and beta-myosin heavy chain, respectively, why did they not also do a Western blot for the "other" isoform as well (Figure 7)? We know that iPSCCM express a mixture, so the relatively mild phenotype that they observe in single knockdown experiments may well be due to concomitant upregulation of the expression of the other isoform. In my point of view this should be checked.

It is likely that in the single knockdown experiments the other isoform is upregulated, which is why we were careful in stating that neither muscle myosin alone is required for sarcomere formation. We do agree this would be an interesting experiment to check beyond the scope of this manuscript.

1. There seems to be a disconnect between the images for myomesin knockdown shown in Figure 8H and the quantification shown in Figure 8I, which makes me wonder whether the image shown in H middle (MYOM1 (1) KD), where the beta-myosin doublets do not seem to be much affected is really representative?

The image shown in the middle of H is representative of the mean length of beta-myosin doublets in MYOM1 (1) KD hiCMs. While the beta-myosin doublets are still present and organized, they are significantly shorter. In the zoomed out image, you can appreciate much shorter arrays of beta-myosin doublets that, while extending across the entire cell, are thinner than control cells.

**Reviewer #2 (Public Review):**
Neininger-Castro et al report on their original study entitled "Independent regulation of Z-lines and M-lines during sarcomere assembly in cardiac myocytes revealed by the automatic image analysis software sarcApp", In this study, the research team developed two software, yoU-Net and sarcApp, that provide new binarization and sarcomere quantification methods. The authors further utilized human induced pluripotent stem cell-derived cardiomyocytes (hiCMs) as their model to verify their software by staining multiple sarcomeric components with and without the treatment of Blebbistatin, a known myosin II activity inhibitor. With the treatment of different Blebbistatin concentrations, the morphology of sarcomeric proteins was disturbed. These disrupted sarcomeric structures were further quantified using sarcApp and the quantification data supported the phenotype. The authors further investigated the roles of muscle myosins in sarcomere assembly by knocking down MYH6, MYH7, or MYOM in hiCMs. The knockdown of these genes did not affect Z-line assembly yet the knockdown of MYOM affected M-line assembly. The authors demonstrated that different muscle myosins participate in sarcomere assembly in different manners.
**Reviewer #3 (Public Review):**
Neininger-Castro and colleagues developed software tools for the quantification of sarcomeres and sarcomere-precursor features in immunostained human induced pluripotent stem cellderived cardiac myocytes (hiCMs). In the first part they used a deep-learning- based model called a U-Net to construct and train a network for binarization of immunostained cardiomyocyte images. They also wrote graphical user interface (GUI) software that will assist other labs in using this approach and made it publicly available. They did not compare their approach to existing ones, but an example from one image suggests their binarization tool outperforms Otsu thresholding binarization.In the second part they developed a software tool called sarcApp that classifies sarcomere structures in the binarized image as a Z-Line or Z-Body and assigns each to either a myofibril or to stress fibers. The tools can then automatically count and measure multiple features (33 per cell and 24 per myofibril) and report them on a per-cell, per-myofibril, and per- stress fiber basis.To test the tools they used Blebbistatin to inhibit sarcomere assembly and showed that the sarcApp tool could capture changes in multiple features such as fewer myofibrils, fewer Z-Lines, decreased myofibril persistence, decreased Z-Line length and altered myofibril orientation in the Blebbistatin treated cells. With some changes the tool was also shown to quantify sarcomeres in titin and myomesin stained cardiomyocytes.Finally they used sarcApp to quantify the changes in sarcomere assembly after siRNA mediated knockout of MYH7, MYH7, or MYOM. The analysis indicates that neither MYH6 nor MYH7 knockdown perturbed the assembly of Z- or M-lines, and that knockdown of MYOM perturbed the A-band/M-Line but not the Z-Line assembly according to features captured by the sarcApp tool.Overall the authors developed and made publicly available an excellent software tool that will be very useful for labs that are interested in studying sarcomere assembly. Multiple features that are difficult to measure or count manually can be automatically measured by the software quickly and accurately.There are however some remaining questions about these tools:1. The binarization tool which is tailored to sarcomere image binarization appears promising but was not systematically compared with existing approaches.

We compared it with the existing approach we used previously in the lab, which was Otsu’s method for binarization. We are not aware of several other binarization approaches to compare to, other than using other machine learning techniques that are less advanced than a U-Net, the current standard in image-to-image translation.

1. How robust is the tool? The tool was tested on images from one type of cardiomyocytes(hiCMs) taken from one lab using Nikon Spinning Disk confocal microscope equipped with Apo TIRF Oil 100X 1.49 NA objective or instant Structured Illumination Microscopy (iSIM), using deconvolution (Microvolution software) and in a specific magnification. It remains to be seen whether the tool would be equally effective with images taken with other microscopy systems, with other cardiomyocytes (chick or neonatal rat), with different magnifications, live imaging, etc.

We tested the software with several magnifications, with live imaging, and with other tissues. We did not include the information in the manuscript because the data we tested the software with is for future manuscripts studying different aspects of sarcomere formation and maintenance. sarcApp reliably identifies Z-Lines and sarcomeres with deconvolved widefield fluorescence images of hiCMs and frozen human tissue, and are currently using it to measure zebrafish data for another study. Further, it works for live imaging with an actinin2-GFP (or similar) label. For the titin quantification, we would recommend using only 60-100X magnification, as the titin structures (doublets and rings) are not resolvable at lower magnifications.

1. The tool was developed for evaluation of sarcomere assembly. The authors show that for this application it can detect the perturbation by Blebbistatin, or knockdown of sarcomeric genes. It remains to be seen if this tool is also useful for assessment of sarcomere structure for other questions beside sarcomere assembly and in other sarcomere pathologies.

While this is beyond the scope of this specific methods paper, we welcome other researchers to use our software for other questions in other pathologies. We are currently doing the same for other manuscripts from our lab.

**Reviewer #1 (Recommendations For The Authors):**
1)"alpha-actinin..., which border the sarcomeric contractile machinery (thin and thick filaments); Z-lines do NOT border thick filaments in a relaxed sarcomere

We have removed “(thin and thick filaments)” from the text.

1. myomesin targeting siRNAs (gene name MYOM): there are actually three genes encoding for myomesin family members, specify, which one was targeted (I am assuming MYOM1).

Thank you for the clarification: we do target MYOM1

1. I am not surprised that they found not many mature Z-lines in the absence of both sarcomeric myosins; a similar codependence of assembly of mature Z-discs and the presence of functional thick filaments was previously shown by Geach and colleagues in 2015 (PMID 25845369)

Thank you for sharing this manuscript: we have added a reference to it in our study.

**Reviewer #2 (Recommendations For The Authors):**
This work offers the possibility to gain more insights into the process of sarcomere assembly through the advancement in sarcomeric or myofibril structure analyses. However, some clarifications are needed from the authors, please see below for the comments.1. It is recommended that the authors include the time points for replating and harvesting hiCMs. After replating, the cardiomyocytes require at least three to four days for sarcomeric structures to reform. If the hiCMs were fixed before sarcomere assembly had completed, the staining of sarcomeric proteins including ACTN2 and titin could be compromised and it is difficult to tell if the phenotypes observed were consequences of drug treatments or knockdown of sarcomeric genes or simply because the replating hiCMs were fixed before their sarcomeric structures had fully regrown. It is also recommended that the authors replate hiCMs at a fixed time point to avoid discrepancies in the data.

Cardiomyocytes do not require three to four days for sarcomeric structures to re-form, and indeed only require 24 hours, with the first sarcomeres typically appearing at ~6 hours. We and others have published several studies demonstrating this (Fenix et al., eLIfe 2018, Taneja, Neininger and Burnette MBoC 2020, Chen et al. Nature Methods, 2022). While sarcomeres continue to develop and turn over after this time, our lab is interested in the beginning steps of sarcomerogenesis rather than the turnover of mature structures.

1. The sarcApp automatically identifies Z-lines and Z-bodies; however, is there an option for the users to set their own thresholds? Some users may select different criterions when quantifying sarcomeres. Moreover, the Z-lines and Z-bodies identified by the software are not always accurate. Can the users modify the list manually in an unbiased way. If this function is not available, the authors may consider adding this function to their software. sarcApp measures Zline and Z-bodies length but does not measure Z-line and Z-bodies width, but sometimes it is also necessary to measure the width.

Absolutely, users can modify the thresholds to identify Z-Lines and Z-Bodies. There is not a way for users to modify the list in an unbiased way per se, as editing the list of Z-Lines and Z-Bodies based on non-mathematical measurements is inherently biased, but the user is free to add in other Z-Lines and Z-Bodies as they wish. In this context, “manually” and “unbiased” is mutually exclusive.

1. It is recommended that the authors include the original images beside the sarcomeric structures identified by sarcApp (Figure 2A, 2C, 4C-F and more). It would be easier to compare the original Z-lines and Z-bodies with those identified by the software.

We have added these in Author response image 1.

**Author response image 1. sa4fig1:** Uncropped images and merges from Figures 2, 4 and 6, respectively.

1. The M-line length quantification data in Figure 3G, 5F, and 6H showed different colored-dots labeling n1 to n3, but the authors did not discuss the significance of these symbols.

We are not sure what the reviewer means by this statement: there is no significance of the different colored dots other than to mark the biological replicate shown. These graphs were created using SuperPlots, which was not stated in the original methods. It has now been added to the Statistical Analysis section.

1. Can the authors elaborate more on the reasons why they treated Blebbistatin at concentrations of 50µM and 100µM. Previous studies showed that 25µM of Blebbistatin was sufficient to delay the transformation of cardiomyocytes (PMID 27072942). Can the authors also comment on why they selected 6 hours, 12 hours, and 24 hours post replating for drug treatment. Moreover, the drug treatment at different time points was only done on ACTN2 but not titin or myomesin.

We selected 6, 12, and 24 hours for actinin2 to show the time course of sarcomere formation and to show that sarcomeres are developed by 24 hours, as also mentioned above. We are interested in future studies of the time course of titin and myomesin over time, and are working on it in the lab.

We chose 50 and 100 µM Blebbistatin as these completely blocked sarcomere assembly whereas treatment with 25 µM did not. This manuscript is a methods paper that aims to validate sarcApp and show how it could be used. We did not intend for it to be a comprehensive study of how different concentrations of blebbistatin affects sarcomere assembly.

We are also unsure what the reviewer means by “transformation of cardiomyocytes”. The manuscript with the PMID of 27072942 does not address this issue. The paper is a “review and analyze readmission data for patients who received a continuous flow left ventricular assist device (LVAD)”. We assume the reviewer is referring to differentiation. The model system we developed and published in eLife in 2018 does not use differentiating iPSC cardiac myocytes. The hiCMs we use are terminally differentiated but still immature, as they are more transcriptionally similar to primary fetal myocytes. As such, they do not maintain their sarcomeres when they removed from the 96 well and plated onto a glass coverslip for highresolution microscopy. These assemble sarcomeres within 24 hours with the sarcomeres forming close to the dorsal membrane and then rearrange overtime (e.g., moving from the top of the cell to the bottom) (Fenix et al., eLife 2018). With that said, we do agree with the reviewer that a study of sarcomere assembly in the context of cardiac myocyte differentiation would be a fascinating direction for future studies, and we think sarcApp could facilitate such studies.

1. The authors mentioned that the myofibrils of Z-line, titin, and M-line were randomly oriented after Blebbistatin treatments. The myofibrils were randomly oriented for titin and M-line. However, the orientation of Z-line after 50µM Blebbistatin treatment was not necessarily random, only the orientation after 100µM Blebbistatin treatment was randomized. The authors might consider changing bar graph to other types of charts if the orientation was really randomized after quantification.

We find that the bar chart is the most informative to us, but users can consider other types of charts in their analyses.

1. It is recommended that the authors include images staining ACTN2 at lower magnifications (Figure 1A, 1C). With current images, it is true that yoU-Net can separate Z-lines from Z-bodies yet it is difficult to tell if yoU-Net can still distinguish Z-lines from Z-bodies with larger images or it only applies to a small portion of the image.

The yoU-Net can distinguish Z-Lines from Z-Bodies with images of any size, as image size (height vs. width in pixels) does not affect how binarization occurs. During binarization, the only pixel requirement is that the width and height are divisible by 8 (for downsampling purposes). Usually this is not the case with raw images, so the image borders are slightly cropped to make them usable. In terms of resolution, we recommend using 60X-100X objectives on confocal or superresolution data for the clearest results. We have, however, successfully binarized deconvolved widefield images at 100X as well.

1. The authors mentioned that the knockdown of MYH7 did not affect Z-lines and M-lines; however, the structures of ACTN2, myomesin, and titin appeared more organized as compared to those in control.

We agree that the sarcomeres and myofibrils look slightly more organized, and did mean to state that the knockdown did not negatively affect Z-Lines and M-Lines and have updated the manuscript to be more accurate.

1. Please provide the merge images for Fig. 4D, 4E, 6B

The merge images for Fig. 4D, 4E, and 6B are included with the original images requested above (point 3)

1. In the text, they described" "antibodies to the titin I-band localize to both MSFs and sarcomeres in hiCMs (Figure 4A). Titin forms ring-like structures around the Z-Bodies of MSFs that are closer to the apparent sarcomere transition point (Figure 4A)" However, based on the antibody information they provided, it is not explicitly recognized for N-or C-terminus TITIN. Please provide TTN N-terminus or TTN-C terminus co-stainings with ACTN2 antibody to understand which part of TTN together with ACTN2 forms a Z-Body.

The TTN antibody is an N-terminal antibody localizing to the I-Band region of sarcomeres. We agree with the reviewer that a more thorough study of titin will be of interest and we are currently undertaking such a study. However, this is a methods paper presenting a tool. While some of the data we present does point to mechanistic hypotheses, it is beyond the scope of this study to fully characterize titin during sarcomere assembly.

1. TITIN doublet was used to indicate a sarcomere in Fig. 4C-D. Moreover, they also used another combination (myomesin and F-ACTIN) to label a sarcomere in Fig. 6D. Can they compare the difference between these two methods or by using these two methods (TITIN doublet) and (myomesin and F-ACTIN), how is the average length of sarcomere? Will the sarcomere length be the same?

We noted in the manuscript that due to the organization of titin doublets (wrapping around the ends of Z-Lines) that the average titin doublet will be approximately 0.3 um longer than the ZLine. We did not expect to see a difference in lengths of myomesin M-Lines and mature actinin2 Z-Lines and indeed do not see major differences in the average lengths (between 2.0 and 2.5 um in 24 hour control cells)

1. They used siRNA method to knockdown MYH6, MYH7 and MYOM and concluded that the knockdown of these genes did not affect the Z-line assembly. Even though they showed very nice knockdown efficiency of these proteins, they should (1) co-stain MYH6/TITIN/actinin2 and MYH6/ myomesin /actinin2 for Fig. 7C. (2) MYH7/TITIN/actinin2 and MYH7/ myomesin /actinin2 for Fig. 7I. (3) MYOM1/TITIN/actinin2 and MYOM2/TITIN/actinin2 for Fig. 8A. (4) MYH7/MYOM1 and MYH7/MYOM2 for Fig. 8H to make sure the cells they measured were truly knockdownpositive cells,

The antibodies for alpha and beta myosin are not very efficient for immunofluorescence, and work best for western blots. We decided also to choose a random subset of the cells on the dish to be sure to eliminate any risk of cherry-picking. While imaging cells on the dish, we looked only at the DAPI nuclear channel and selected 50 cells minimum per dish with only this channel, then imaged the other channels.

Minor comments:1. Well-organized sarcomere structure on DMSO treated cells in Fig.5A and Fig. 6A, but it was disarray in Fig. S3M. Why?

Figure S3 shows hiCMs that have only been allowed to spread for 6 hours, which have not formed mature sarcomeres yet, hence the disarray.

1. Fig 1A, Fig2B: please label the name of the antibody, not the actin filament

We used phalloidin labelling here, which marks actin filaments. We have updated the figure legends to be more clear. Thank you!

1. Fig. 7I: actinin2 instead of actinin

Thank you for catching this! We have fixed it.

**Reviewer #3 (Recommendations For The Authors):**
Testing the app using images shot by other microscopy systems, magnifications, and cardiomyocytes from other species, as noted in the public review above, should make the app even more wildly useful.A more formal head-to-head comparison with other approaches will be more convincing in showing the new tool is superiorI also think that a more detailed protocol for using the app will help other investigators.The app counts and measures many features, but it is not always clear how and using what algorithm these are measured. Including these details in a protocol or even as comments in the code will be very helpful for others.

The protocol found on the public GitHub for the app will help other investigators to download, use, and understand the application. We have received contact from researchers who have been able to use the application without assistance from us, which is a good sign that the application is user-friendly and that the online protocol is sufficient.